# C-type lectin receptor 2d forms homodimers and heterodimers with TLR2 to negatively regulate IRF5-mediated antifungal immunity

Fan Li[1,2,3,4], Hui Wang[2,3,4], Yan-Qi Li[2,3], Yebo Gu ®[1] ✉ & Xin-Ming Jia ®[2,3] ✉

Dimerization of C-type lectin receptors (CLRs) or Toll-like receptors (TLRs) can alter their ligand binding ability, thereby modulating immune responses. However, the possibilities and roles of dimerization between CLRs and TLRs remain unclear. Here we show that C-type lectin receptor-2d (CLEC2D) forms homodimers, as well as heterodimers with TLR2. Quantitative ligand binding assays reveal that both CLEC2D homodimers and CLEC2D/TLR2 heterodimers have a higher binding ability to fungi-derived β-glucans than TLR2 homo-dimers. Moreover, homo- or hetero-dimeric CLEC2D mediates β-glucan-induced ubiquitination and degradation of MyD88 to inhibit the activation of transcription factor IRF5 and subsequent IL-12 production. *Clec2d*-deficient female mice are resistant to infection with *Candida albicans*, a human fungal pathogen, owing to the increase of IL-12 production and subsequent generation of IFN-γ-producing NK cells. Together, these data indicate that CLEC2D forms homodimers or heterodimers with TLR2, which negatively regulate antifungal immunity through suppression of IRF5-mediated IL-12 production. These homo- and hetero-dimers of CLEC2D and TLR2 provide an example of receptor dimerization to regulate host innate immunity against microbial infections.

The innate immune defenses against various microbial infections are initiated by the germ-line encoded pattern recognition receptors (PRRs) that recognize pathogen-associated molecular patterns (PAMPs). The emerging concept in the field is that PRRs can form homodimers and heterodimers, leading to changes in classical features including ligand binding affinity and diversity or receptor activation and signaling. Thus, PRR dimerization may be one of the first lines to regulate innate immune responses.

Toll-like receptors (TLRs) are generally considered to be the principal PRRs involved in microbial recognition and intracellular signaling. In mammals, most TLRs bind various ligands by forming homodimers to induce subsequent immune responses. For example, TLR2 recognizes lipoteichoic acid (LTA) and peptidoglycan (PGN), as

well as TLR3 recognizes double-stranded RNA, both by forming homodimers[1,2]. However, TLR2 heterodimerizes with either TLR1 or TLR6 for efficient ligand binding and discrimination of triacyl and diacyl lipopeptides from bacteria[3,4]. Moreover, synthetic tri-palmitoylated bacterial lipopeptide analogue (Pam3CSK4) and di-acylated macrophage-activating lipopeptide 2 (MALP-2) are highly potent agonists for human TLR2/TLR1 and TLR2/TLR6 heterodimers, respectively[5]. Thus, TLR2 has been shown to detect a wide array of PAMPs through forming homo- or hetero-dimers with other TLRs.

Other 'non-TLR' receptors may also play functions in a dimeric manner. Among these molecules, some C-type lectin receptors (CLRs) bind various ligands by forming homodimers to elicit an appropriate immune response. For example, Dectin-1 and LOX-1 form functional

[1]Department of Stomatology, Shanghai Tenth People's Hospital, School of Medicine, Tongji University, Shanghai 200072, China. [2]Clinical Medicine Scientific and Technical Innovation Center, Shanghai Tenth People's Hospital, Tongji University School of Medicine, Shanghai 200072, China. [3]Key Laboratory of Pathogen-Host Interactions of the Ministry of Education of China, Tongji University, Shanghai 200092, China. [4]These authors contributed equally: Fan Li, Hui Wang. ✉e-mail: abbyyebogu@tongji.edu.cn; jiaxm@tongji.edu.cn

homodimers to recognize fungal β-glucans and oxidized low-density lipoprotein (oxLDL), respectively[6–8]. Our previous study has shown that Dectin-3 and Dectin-2 form heterodimers to bind fungal α-mannans more effectively, thereby leading to potent inflammatory responses against fungal infections with *Candida albicans*, a major human fungal pathogen[9]. Moreover, a recent study shows that CLEC5A, DC-SIGN, and mannose receptor (MR) form a multivalent hetero-complex upon engagement with Dengue virus (DV), where DC-SIGN/MR binds DV with high avidity to facilitate CLEC5A-mediated innate responses against DV infection[10]. Other studies suggest that DV could activate CLEC2 via binding to a DC-SIGN/CLEC2 hetero-multivalent receptor complex in platelets, despite that CLEC2 has low binding affinity to DV[11,12]. Of note, two recent studies show that co-activation of multiple CLEC5A- and TLR2-mediated pathways are required for optimal host innate immunity against infection with DV or *Listeria monocytogenes*[11,13]. In addition, TLR2 forms the complex with non-TLR molecules such as CD36 and Dectin-1. CD36, a member of the scavenger receptor type B family, has a role as a co-receptor for dia-cylglyceride recognition by TLR2/TLR6 heterodimers[14]. Of note, TLR2 co-operates with Dectin-1 to recognize fungal β-glucans for synergistically mediating cytokine production against *C. albicans* infection[15–17]. Thus, it is worth noting that there are some PRRs that co-operate cross-family to regulate immune responses. However, it remains unclear about the possibility and role of heterodimerization between TLRs and CLRs.

As a C-type lectin-like protein, CLEC2D (C-type lectin receptor-2d, also known as LLT1), was originally characterized as the ligand for the orphan CLR CD161 (also known as NKR-P1A) expressed on human NK cells and T cells[18]. CLEC2D has been found to be expressed on tumor cells, infiltrating bone marrow cells, monocytes, natural killer (NK) cells, and subsets of T cells[19]. Previous research has predominantly concentrated on investigating the influence of CLEC2d-CD161 interaction in tumor and anti-viral immunity[19,20], which inhibits NK cell-mediated cytotoxicity and IFN-γ secretion. *Lai* et al. expanded the role of CLEC2D serving as a sensor for cell death to detect histones released from necrotic cells[21]. CLEC2D has been found to carry histone-bound DNA complexes into endosomes, stimulating TLR9-mediated pro-inflammatory responses in macrophages and microglia[21,22]. Our recent findings demonstrate that CLEC2D recognizes glucuronoxylomannan from *Cryptococcus neoformans* to promote the p38-mediated production of arginase-1, leading to the induction of immunosuppressive activity of neutrophilic myeloid-derived suppressor cells (MDSCs), and subsequently inhibiting T cell-mediated anti-fungal responses[23].

In the present study, we performed bimolecular fluorescence complementation assays and found that CLEC2D formed homodimers by itself or heterodimers with TLR2. Furthermore, we showed that both homo- and heterodimeric CLEC2D had high binding abilities to β-glucans, and CLEC2D conferred TLR2 high binding ability to β-glucans through forming heterodimers. More importantly, CLEC2D formed homodimers or heterodimers with TLR2 to negatively regulate IRF5-mediated antifungal immunity against *C. albicans* infection. Together, our data establish a new example in which CLEC2D and TLR2 on immune cells can form homo- or hetero-dimers to negatively regulate antifungal immunity against *C. albicans* infection.

## Results

### CLEC2D forms homodimers by itself or heterodimers with TLR2

To explore whether CLEC2D, Dectin-1 and TLR2 directly interacted with each other, we co-expressed CLEC2D-Flag, Dectin-1-HA and TLR2-Myc in different combinations in 293T cells for co-immunoprecipitation and immunoblot-based analyses (Fig. 1a, b and Supplementary Fig. 1a, b). Dectin-1 was found to have no interaction with TLR2 (Supplementary Fig. 1a). However, we surprisingly observed that CLEC2D directly interacted with TLR2 after stimulation with or without heat-activated *C. albicans* yeast containing β-glucans, but no

interaction with Dectin-1 (Fig. 1a, b and Supplementary Fig. 1b). Moreover, the accumulation of endogenous CLEC2D-associated TLR2 protein in bone marrow-derived dendritic cells (BMDCs) was significantly increased after stimulation with β-glucan-containing particle curdlan or heat-activated *C. albicans* yeast (Fig.1c, d). We further observed the interaction of TLR2 and CLEC2D-Dectin-1 chimeras containing the extracellular domains of CLEC2D when co-expressed in 293T transfectants (Supplementary Fig. 1c). More importantly, the deletion of extracellular domains of either CLEC2D or TLR2 completely impaired their associations whereas the deletion of intracellular or transmembrane domains of CLEC2D or TLR2 had no influence on their associations (Fig. 1e, f). We further generated soluble protein chimeras containing different extracellular fragments of CLEC2D, and found that the removal of 80–99aa of CLEC2D completely impaired its interaction with TLR2 (Fig. 1g). These data indicated that CLEC2D and TLR2 directedly associated with each other, depending on their extracellular domains.

For examining the dimerization of CLEC2D, especially whether it can form heterodimers with TLR2, we performed bimolecular fluorescence complementation (BiFC) assays as described in our previous study[9]. Specifically, we fused CLEC2D and TLR2 to the N- or C-terminal half of yellow fluorescent protein (YFP$^N$ or YFP$^C$), and the physical interaction between CLEC2D and TLR2 or their individual expression in 239 T cells allowed YFP$^N$ and YFP$^C$ to come together and reconstitute a functional YFP, resulting in fluorescence emission. We found that the combination of YFP$^N$-CLEC2D and YFP$^C$-CLEC2D formed complexes on the cell surface and fluorescence was emitted (Fig. 1h and Supplementary Fig. 1d), suggesting that CLEC2D can form homodimers. Importantly, when YFP$^N$-CLEC2D was combined with YFP$^C$-TLR2, complex formation was also observed on the cell surface, accompanied by fluorescence emission (Fig. 1h and Supplementary Fig. 1d), implying that CLEC2D can also form heterodimers with TLR2. To confirm the specificity of CLEC2D homodimer and CLEC2D/TLR2 heterodimer formation, we co-expressed HA-tagged CLEC2D or Myc-tagged TLR2 with the YFP$^+$ CLEC2D homodimers or YFP$^+$ CLEC2D/TLR2 heterodimers in 293T cells. Expression of either CLEC2D or TLR2 competitively reduced the frequency of YFP$^+$ cells expressing homodimers or heterodimers (Supplementary Fig. 1e). In addition, we observed that stimulation with *C. albicans* yeast did not impact the frequency of YFP$^+$ CLEC2D homodimers or YFP$^+$ CLEC2D/TLR2 heterodimers when overexpressed in 293 T cells (Supplementary Fig. 1f). Moreover, the interaction of endogenous CLEC2D and TLR2 in human peripheral blood mononuclear cells (PBMCs) was enhanced after stimulation with β-glucans, which was visualized by immuno-fluorescence microscopy (Supplementary Fig. 1g). Using Fluorescence Resonance Energy Transfer (FRET), we conducted further investigations into the interaction between endogenous CLEC2D and TLR2 in human peripheral blood mononuclear cells (PBMCs). Antibodies specific to human CLEC2D and TLR2 were labeled with the fluorescence donor Cy3 and acceptor Cy5, respectively. Following stimulation with β-glucans, we observed a significant increase in FRET efficiency, indicating a rapid association between CLEC2D and TLR2 on the PBMC membranes (Fig. 1i and Supplementary Fig. 1h). Together, these data implied that CLEC2D formed homodimers by itself or heterodimers with TLR2.

### Both CLEC2D/TLR2 heterodimers and CLEC2D homodimers have high binding capability to fungal β-glucans

To clarify the recognition of *C. albicans* by CLEC2D, we generated a soluble chimera CLEC2D-Fc as described in our recent study[23]. We found that the C-type lectin-like domain of CLEC2D had strong binding ability to *C. albicans* yeast cells (Fig. 2a). Notably, we observed that pre-treatment with established ligands of CLEC2D including dextran sulfate and λ-carrageenan resulted in complete inhibition of their binding (Fig. 2a). However, it is important to note that dextran did not exhibit

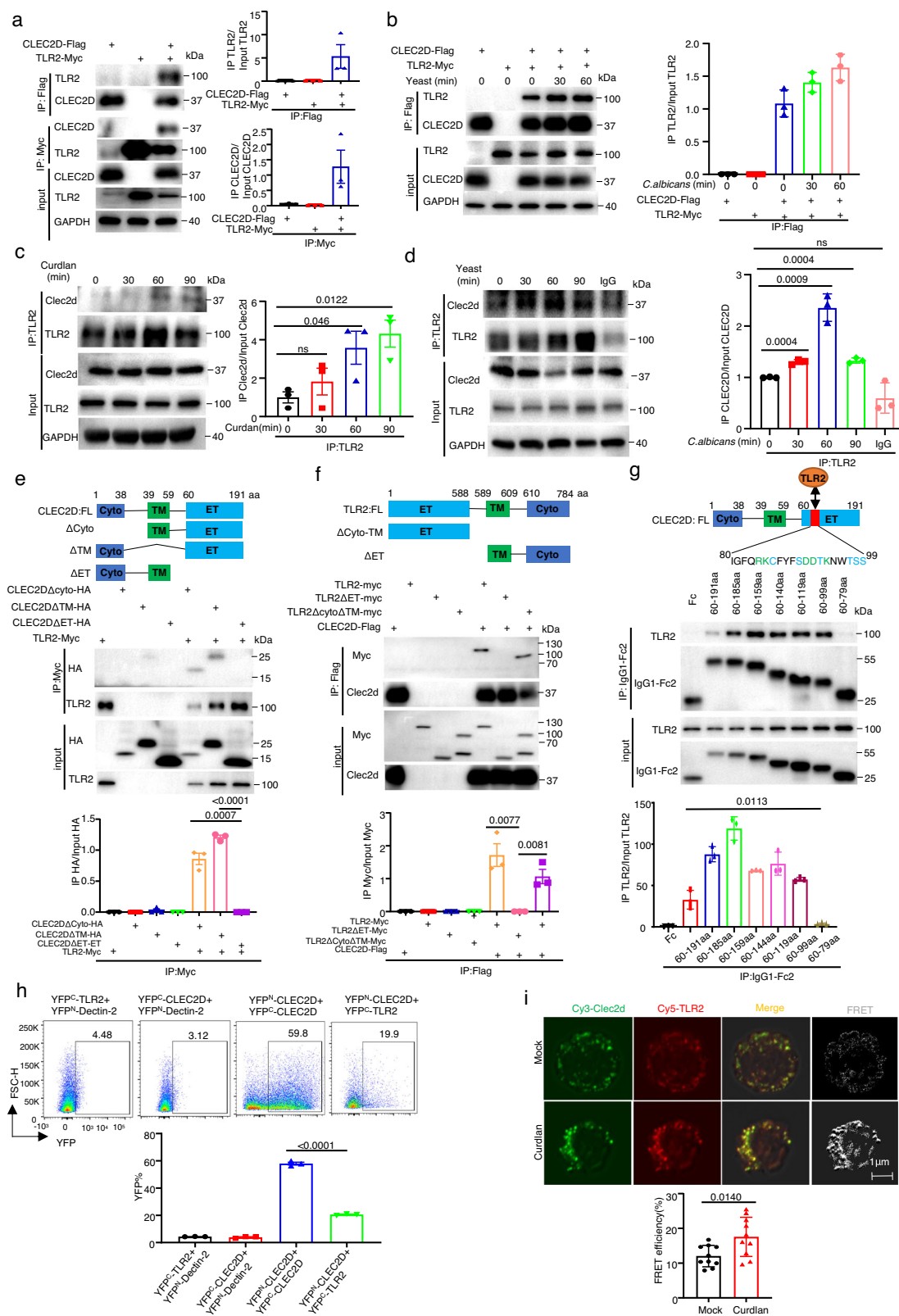

this blocking effect (Fig. 2a), hence reinforcing the specificity of the ligand-receptor interaction. Immunofluorescence microscopy was employed to visualize the binding of β-glucans with CLEC2D or Dectin-1 as a positive control[24]. The results showed consistent overlaps of CLEC2D or Dectin-1 with β-glucans on the surface of *C. albicans* yeast cells (Fig. 2b). In addition, the substitution of positively charged residues, such as Arg180, Lys181, and Lys186, with glutamic acids resulted

in a complete loss of CLEC2D binding to *C. albicans* yeast cells (Supplementary Fig. 2a). Conversely, mutations at Lys169, Arg175, His176, Arg180, and His190 did not significantly affect the binding capability of CLEC2D (Supplementary Fig. 2a). We further confirmed that CLEC2D could directly bind to fungal β-glucan-rich particle curdlan and α-mannans (Fig. 2c). Moreover, we observed that mutations at specific amino acids, namely His190, Arg175, and His176, had a partial impact

**Fig. 1 | CLEC2D forms homodimers by itself or heterodimers with TLR2. a, b** Left panel: Association analyses of human CLEC2D-Flag with human TLR2-Myc expressed in HEK293T cells, which were stimulated without (**a**) or with (**b**) *C. albicans* yeast (MOI = 5) for the indicated time. Right panel: Quantitative analysis of IP TLR2 normalized to input TLR2 and IP CLEC2D normalized to input CLEC2D(**a**) and IP TLR2 normalized to input TLR2 (**b**). **c, d** Left panel: Association analyses of the Clec2d and TLR2 in BMDC stimulated with Curdlan (**c**) and *C. albicans* yeast (**d**) for the indicated time. Right panel: Quantitative analysis of IP Clec2d normalized to input Clec2d (**c, d**). **e, f** Upper panel: Association analyses of ET, TM or Cyto domains of human TLR2-Myc and human CLEC2D-Flag, which were expressed in HEK293T cells as the indicated combinations. Lower panel: Quantitative analysis of IP HA normalized to input HA (**e**) and IP Myc normalized to input Myc (**f**). **g** Upper panel:

Association analyses of different human CLEC2D-Fc truncated fragments with human TLR2-Myc expressed in HEK293T cells. Lower panel: Quantitative analysis of IP TLR2 normalized to input TLR2. **h** Bimolecular fluorescence complementation (BiFC) assay for the frequency of homodimerization of YFP$^N$-CLEC2D with YFP$^C$-CLEC2D or heterodimerization of YFP$^N$-CLEC2D with YFP$^C$-TLR2. **i** The interaction between CLEC2D labeled with Cy3 (green) and TLR2 labeled with Cy5 (red) in human PBMC (1 × 10$^6$ cells/well) as assessed by FRET after 60 min stimulation with curdlan (20 μg/well). Scar bars = 1μm. Cyto cytoplasma, TM transmembrane, ET Extracellular. Data were presented as mean ± SEM; *n* = 3 (**a**–**h**), *n* = 10 (**i**) biologically independent samples. Data were analyzed by unpaired two-sided Student's t-test in (**a**–**i**). Source data are provided as a Source Data file.

on the binding of CLEC2D to β-glucans and α-mannans (Fig. 2c). Conversely, mutations at Arg180, Lys181, Lys186, and Lys169 of CLEC2D completely abolished its ability to bind to these ligands (Fig. 2c). These findings suggested that CLEC2D had a specific recognition of fungal β-glucans and α-mannans, utilizing its positively charged residues.

In view of our evidence that CLEC2D could be induced to interact with TLR2 upon stimulation with β-glucans or *C. albicans* yeast, we examined the ligand-binding abilities of homodimers or heterodimers of CLEC2D and TLR2 to fungi-derived β-glucans and α-mannans or Pam3CSK4, a well-characterized TLR2 ligand. To this end, we performed ELISA assay to examine the binding abilities to the extracellular domain of Fc-fused CLEC2D or His-fused TLR2, which were used alone or in combination. HRP-labelled anti-Fc or -His antibodies were employed for detection, and it is important to note that these antibodies did not cross-react with the His-TLR2 or Fc-CLEC2D proteins (Supplementary Fig. 2b). CLEC2D/TLR2 heterodimers exhibited similar binding abilities to β-glucans as compared to CLEC2D homodimers (Fig. 2d). In addition, both CLEC2D/TLR2 heterodimers and CLEC2D homodimers displayed higher binding ability to β-glucans compared to TLR2 homodimers (Fig. 2d). However, when binding to α-mannans or Pam3CSK4, the binding abilities of homo- or heterodimeric complexes containing the extracellular domains of CLEC2D and TLR2, either alone or in combination, were comparably low (Fig. 2d). Interestingly, it was observed that dilutions of TLR2 protein did not affect the binding ability of its heterodimerized form with CLEC2D (Fig. 2e). On the other hand, the binding ability of heterodimerized TLR2 with CLEC2D was significantly reduced by dilutions of CLEC2D protein (Fig. 2e). Furthermore, while CLEC2D protein dilutions slightly decreased the binding ability of its heterodimerized form with TLR2 for α-mannans or Pam3CSK4, multiple dilutions of TLR2 protein had no effect on the binding ability of its heterodimerized form with CLEC2D for α-mannans or Pam3CSK4 (Fig. 2e). These findings provided valuable insights into the ligand-binding abilities of CLEC2D homodimers and heterodimers with TLR2 for various fungal-derived molecules, highlighting their potential role in recognizing and responding to fungi.

In order to further investigate the binding interactions, we utilized His-TLR2 as a probe to assess its binding frequency to *C. albicans* yeast cells. We performed flow cytometry analysis, both with and without the addition of Fc-CLEC2D or Fc-Dectin-1 (Fig. 2f). Our findings demonstrated that the inclusion of Fc-CLEC2D significantly enhanced the frequency of TLR2 binding to *C. albicans* yeast cells (Fig. 2f). Conversely, the presence of Fc-Dectin-1 competitively reduced the binding frequency of the heterodimeric TLR2/CLEC2D (Fig. 2f). Notably, the addition or dilution of His-Dectin-1 protein did not significantly impact the binding ability of Fc-CLEC2D to β-glucans or the binding frequency of Fc-CLEC2D to *C. albicans* yeast cells (Supplementary Fig. 2c, d). These observations collectively suggested that both CLEC2D/TLR2 heterodimers and CLEC2D homodimers exhibited a high binding ability to fungal β-glucans.

## CLEC2D/TLR2 heterodimers and CLEC2D homodimers promote MyD88 ubiquitination and degradation to inhibit β-glucan-induced IRF5 activation

To explore the innate immune responses mediated by homo- and heterodimers of CLEC2D, we stimulated BMDCs from wild-type and *Clec2d*-deficient mice with β-glucans, α-mannans or Pam3CSK4. Surprisingly, we observed that the absence of CLEC2D in BMDCs significantly enhanced the phosphorylation and nuclear translocation of IRF5 in response to β-glucan stimulation (Fig. 3a, b). However, we did not observe any significant effect of *Clec2d* deficiency on the phosphorylation of Syk and p65 in BMDCs (Fig. 3a, b). Furthermore, in *Clec2d*-deficient BMDCs, we found that the phosphorylation of IRF5 significantly increased upon stimulation with *C. albicans* yeast, but not with hyphae (Supplementary Fig. 3a, b). In addition, we conducted experiments using RAW264.7 cells overexpressing CLEC2D. We observed that the phosphorylation of IRF5 induced by β-glucan stimulation was significantly inhibited in these cells, while the phosphorylation of p65 remained unaffected (Fig. 3c, d). However, *Clec2d* deficiency slightly increased β-glucan-induced phosphorylation of p38, JNK and ERK1/2, which were not affected by the overexpression of Clec2d in RAW264.7 cells (Supplementary Fig. 3c, d). Furthermore, stimulation with α-mannans or Pam3CSK4 did not significantly influence the phosphorylation levels of IRF5, Syk, p38, JNK, ERK1/2 and p65 stimulated with α-mannans or Pam3CSK4 between wild-type and *Clec2d*-deficient BMDCs (Supplementary Fig. 3e, f). These findings suggested that CLEC2D negatively regulated the activation of IRF5 specifically induced by β-glucans, but not by α-mannans or Pam3CSK4.

To unravel the underlying mechanisms that CLEC2D negatively regulated IRF5 activation, we examined whether *Clec2d* deficiency or overexpression affected MyD88 degradation, which is a critical regulatory step for IRF5 activation[25]. Surprisingly, our findings revealed that the degradation of MyD88 in BMDCs was promoted in a time-dependent manner upon stimulation with β-glucans or *C. albicans* yeast (Fig. 3e and Supplementary Fig. 3a). Conversely, *Clec2d* deficiency was found to stabilize MyD88 (Fig. 3e and Supplementary Fig. 3a), indicating a preference for its degradation. Furthermore, the overexpression of CLEDC2D in RAW264.7 cells resulted in a time-dependent increase in the degradation of MyD88 upon β-glucan stimulation (Fig. 3f). This degradation of MyD88 was completely inhibited when the cells were pretreated with the proteasome inhibitor MG132 (Fig. 3f). These findings strongly suggested that CLEC2D recognized β-glucans and subsequently promoted the degradation of MyD88 through a proteasome-mediated pathway. We further found that the MyD88 inhibitor TJ-M2010-530[26] effectively blocked the phosphorylation of IRF5 mediated by MyD88 in response to β-glucan stimulation (Fig. 3g). Thus, we could conclude that CLEC2D facilitated the degradation of MyD88, thereby inhibiting β-glucan-induced IRF5 activation.

Additionally, we generated double knockout (dKO) mice lacking both *Clec2d* and *TLR2* (Supplementary Fig. 3g). Interestingly, we observed that the absence of either *Clec2d* alone or in combination

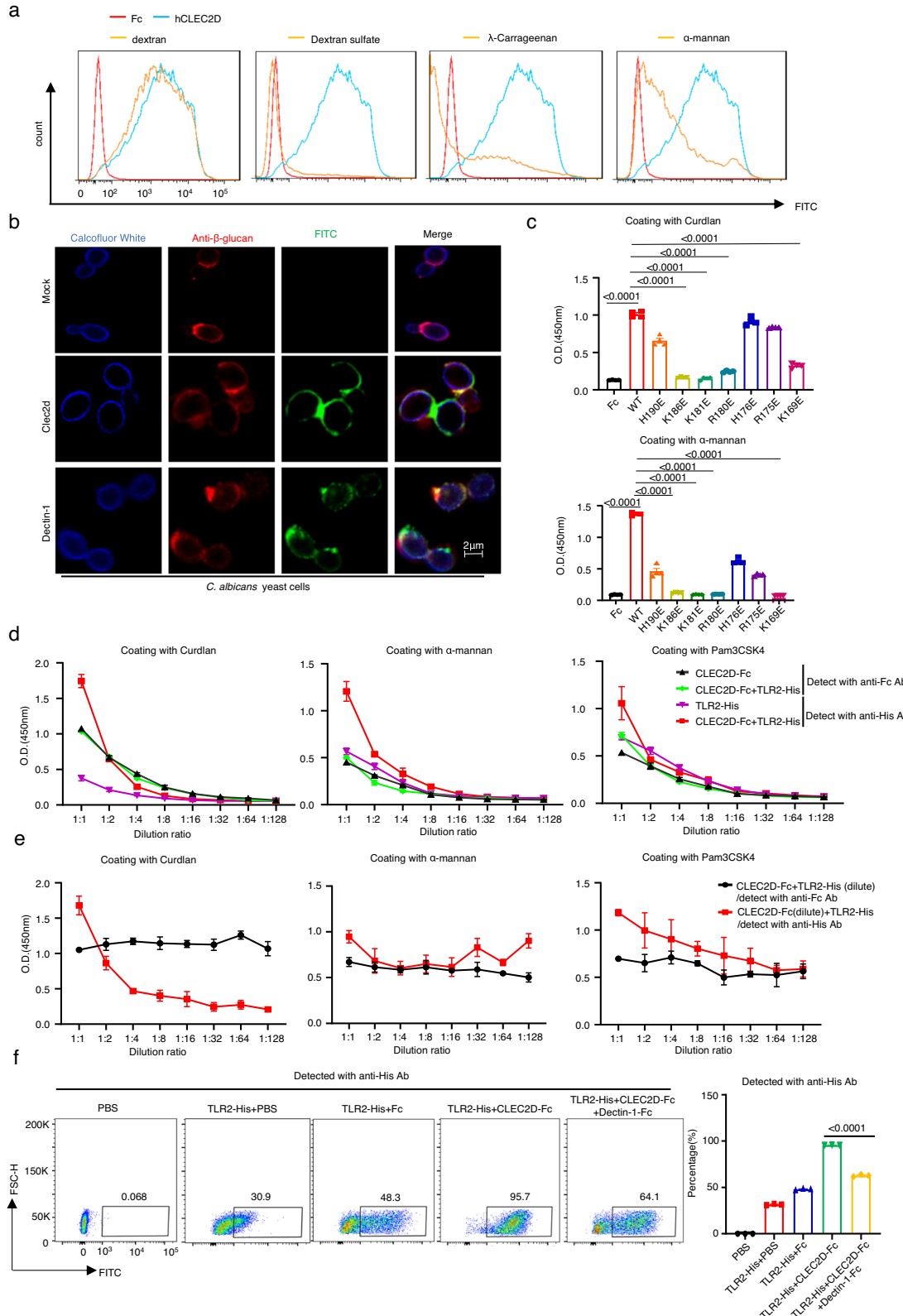

with *TLR2* in BMDCs significantly enhanced β-glucan-induced IRF5 phosphorylation and MyD88 stabilization (Fig. 3h). In contrast, TLR2 deficiency alone had no effect on IRF5 activation and MyD88 degradation (Fig. 3h). Thus, these data implied that CLEC2D/TLR2 heterodimers or CLEC2D-homodimers, but not TLR2 homodimers, promoted β-glucan-induced MyD88 degradation and subsequently inhibited IRF5 activation.

## CLEC2D promotes Smurf1-mediated degradation of MyD88 through K48-linked ubiquitination

We conducted further experiments to explore whether CLEC2D could mediate the ubiquitination of MyD88. We employed the tandem-repeated ubiquitin-binding entity (TUBE) ubiquitination assay and observed that MyD88 ubiquitination significantly increased upon stimulation with β-glucans (Fig. 4a). However, the ubiquitination of

**Fig. 2 | Both CLEC2D/TLR2 heterodimers and CLEC2D homodimers have high binding affinity to fungal β-glucans. a** Flow cytometry assay of the competitive binding of *C. albicans* yeast cells with CLEC2D-hIgG1-Fc or hIgG1-Fc (hFc) using the indicated polysaccharides. **b** Representative immunofluorescent staining assay of the binding of CLEC2D-hIgG1-Fc (green) or Dectin-1-hIgG1-Fc (green) with *C. albicans*. Chitin was stained with Calcofluor White (50 µg/ml, blue), and β-glucan was stained with anti β-1,3-glucan antibody (red). Scar bars = 2 µm. Three times experiments were repeated independently with similar results. **c** ELISA assay for the binding of indicated amino acid point mutated proteins of CLEC2D-hIgG1-Fc or hIgG1-Fc (hFc) with plate-coated curdlan or α-mannan (5 µg/well). **d** ELISA results for the binding of diluted CLEC2D-Fc and TLR2-His homo- or heterodimers with plated-coated curdlan, α-mannan or Pam3CSK4 (1 µg/well), which were detected with anti-Fc or -His Abs, respectively. **e** ELISA results for the binding of CLEC2D-Fc/TLR2-His heterodimers containing the dilution of either CLEC2D-Fc (Red line) or TLR2-His (Black line) with plated-coated curdlan, α-mannan or Pam3CSK4 (1 µg/well), which were detected with anti-Fc or -His Abs, respectively. **f** Flow cytometry assay of the competitive binding of Dectin-1-hIgG1-Fc with the CLEC2D-Fc/TLR2-His heterodimers to *C. albicans* yeast cells, which was detected with anti-His Ab. Data were presented as mean ± SEM; *n* = 3 (**d**–**f**), *n* = 4 (**c**) biologically independent samples. Data were analyzed by unpaired two-sided Student's *t*-test in (**c**–**f**). Source data are provided as a Source Data file.

MyD88 was dramatically impaired in BMDCs lacking CLEC2D (Fig. 4a). In addition, the exogenous expression of CLEC2D in 293T transfectants effectively promoted MyD88 ubiquitination (Fig. 4b). To understand the underlying molecular mechanism of CLEC2D-mediated degradation of MyD88, we transfected vectors expressing HA-tagged Lys48 (K48)- or K63-linked ubiquitin into 293T cells. We discovered that the overexpression of CLEC2D enhanced K48-linked ubiquitination of MyD88 rather than K63-linked ubiquitination (Fig. 4c). Consequently, the overexpressed CLEC2D in 293T cells facilitated the degradation of MyD88 through K48-linked ubiquitination (Fig. 4d, e).

Previous studies have demonstrated that the E3 ligases Smurf1 or SPOP can mediate the ubiquitination and subsequent proteasomal degradation of MyD88 in TLR signaling pathways[27,28]. In our investigation, we found that the inhibition of Smurf1 with its inhibitor Smurf1-IN-A01 completely blocked Clec2d-mediated degradation of MyD88 in RAW264.7 cells upon stimulation with β-glucans (Fig. 4f). In contrast, the inhibition of SPOP with its inhibitor SPOP-IN-6b had no effect on MyD88 degradation (Fig. 4f). Based on these findings, we could infer that CLEC2D promoted Smurf1-mediated proteasomal degradation of MyD88, thereby inhibiting β-glucan-induced IRF5 activation.

## CLEC2D/TLR2 heterodimers and CLEC2D homodimers negatively regulate IRF5-mediated IL-12 production

To further investigate the role of CLEC2D in innate immune responses, we conducted an experiment where we stimulated wild-type and Clec2d-deficient bone BMDCs with β-glucans for 3 h. Subsequently, we performed microarray analysis to identify differentially expressed genes (DEGs). Gene Ontology (GO) analysis revealed that out of the 311 DEGs, many were closely related to CLR and TLR signaling pathways (Fig. 5a and Supplementary Fig. 4a). Interestingly, we observed that *Clec2d*-deficient BMDCs exhibited higher expression of *Il12a* (encoding IL-12p35 subunit) and *Il12b* (encoding IL-12p40 subunit) compared to wild-type BMDCs after β-glucan stimulation (Fig. 5b). To validate these findings, we performed quantitative real-time polymerase chain reaction (PCR) analysis. Our results demonstrated that *Clec2d* deficiency led to a significant increase in the β-glucan-induced expression levels of *Il12a*, while the expression of *Il12b* remained unaffected (Fig. 5c). To further investigate the role of CLEC2D, we overexpressed CLEC2D in RAW264.7 cells and observed a significant reduction in the β-glucan-induced expression of *Il12a*, while *Il12b* expression was unchanged (Fig. 5d).

It has been observed that the secretion of IL-12 by DCs and macrophages requires prior priming with IFN-γ and subsequent stimulation with LPS[29]. In our direct ex vivo assay, we discovered that deficiency of *Clec2d* resulted in a notable increase in the production of IL-12 induced by β-glucan in IFN-γ-primed BMDCs, along with a slight enhancement in the production of IL-12p40 and IL-23 (Fig. 5e and Supplementary Fig. 4b). To understand the underlying mechanism, we further conducted chromatin immunoprecipitation (ChIP) assays to assess the impact of *Clec2d* deficiency on the binding of IRF5 to the *Il12* promoter. IRF5 is known to regulate the transcription of *Il12a* and *Il12b*[30]. As expected, we observed a significant increase in the binding

of β-glucan-induced IRF5 to the *Il12a* promoter in BMDCs lacking CLEC2D (Fig. 5f). Additionally, a slight increase was observed in the binding of IRF5 to the promoter of *Il12b* (Fig. 5f). These findings suggest that CLEC2D plays a crucial role in negatively regulating the expression of *Il12a* in response to β-glucan stimulation. Moreover, inhibition of either IRF5 or Myd88 activation with their respective inhibitory N5-1[31] or TJ-M2010-5 completely blocked β-glucan-induced IL-12 production in IFN-γ primed wild-type and *Clec2d*-deficient BDMCs (Fig. 5g, h). These data suggested that CLEC2D negatively regulated Myd88/IRF5-mediated IL-12 production.

More importantly, in BMDCs with IFN-γ priming and β-glucan stimulation, the deficiency of either *Clec2d* alone or in combination with *TLR2* deficiency resulted in a significant increase in IL-12 production, whereas TLR2 deficiency alone in BMDCs completely blocked it (Fig. 5i). However, the deficiency of either Clec2d and TLR2 alone or together had minimal or no effect on β-glucan-induced IL-12p40 production, respectively (Fig. 5i). Moreover, in BMDCs with or without IFN-γ priming, the deficiency of either *TLR2* alone or in combination with *Clec2d* completely blocked Pam3CSK4-induced production of IL-12 and IL-12p40 (Fig. 5i). Together, these data implied that CLEC2D/TLR2 heterodimers and CLEC2D homodimers negatively regulated β-glucan-induced IL-12 production.

## *Clec2d* knockout mice are highly resistant to infection with *C. albicans* through increasing IL-12 production

To examine the effect of CLEC2D-mediated immune responses on the host's ability to clear systemic *C. albicans* infections, we challenged wild-type or *Clec2d*-deficient mice with a sublethal dose ($2 \times 10^5$) of *C. albicans* strain SC5314. We showed that *Clec2d* deficiency led to a significant improvement in the survival rate of the mice ($P < 0.01$; Fig. 6a). Consistently, the fungal burdens in the kidneys and brains of *Clec2d*-deficient mice were significantly lower than those in wild-type mice after infection for 2 and 4 days (Fig. 6b). Histological analyses using periodic acid-Schiff staining showed minimal fungal presence in the kidneys of *Clec2d*-deficient mice, while wild-type mice exhibited many hyphae (Fig. 6c). Histological analysis with hematoxylin-eosin staining also showed that kidney damage was alleviated in *Clec2d*-deficient mice compared with those in wild-type mice (Fig. 6c). Furthermore, *Clec2d* deficiency resulted in a notable upregulation of IL-12p70, IL-12p40 and IFN-γ production on Day 1 following *C. albicans* infection *s* (Fig. 6d). However, it did not have any impact on the production of TNF-α, IL-6, and IL-1β (Supplementary Fig. 5a). These data implied that CLEC2D exerted a negative regulatory effect on the production of IL-12p70, IL-12p40 and IFN-γ, thereby suppressing the host immune response against *C. albicans* infection.

It is well-established that IFN-γ is produced by gamma delta (γδ) T and NK cells in response to IL-12 during the early stages of infection[32]. We observed a higher number and frequency of NK cells, particularly those producing IFN-γ, in the kidneys of *Clec2d*-deficient mice compared to wild-type mice on Day 1 and 2 after infection with *C. albicans* (Fig. 6e, f and Supplementary Fig. 5b). However, no significant difference was observed in the quantity or frequency of γδT cells (Supplementary Fig. 5c).

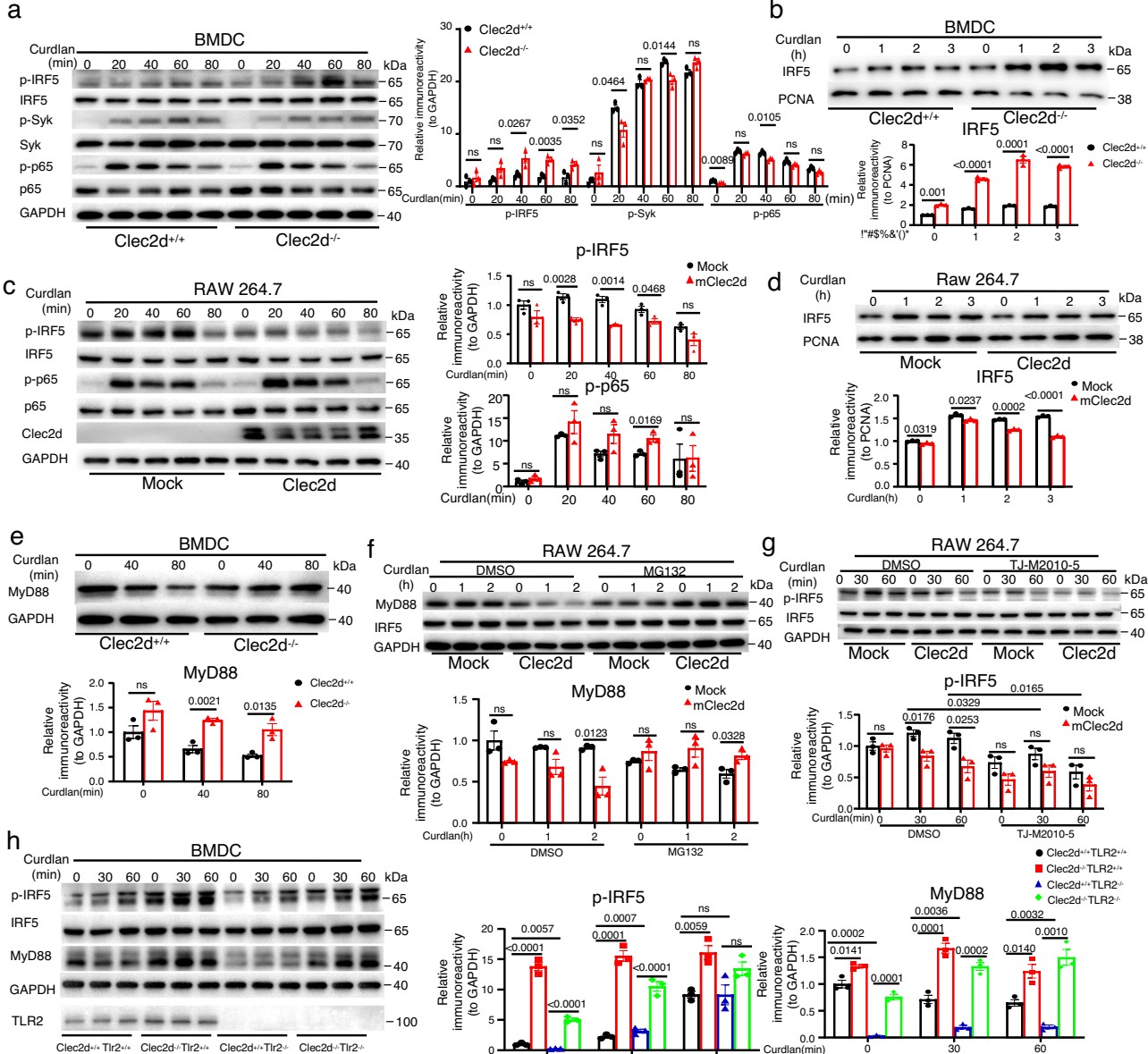

**Fig. 3 | CLEC2D/TLR2 heterodimers and CLEC2D homodimers promote MyD88 ubiquitination and degradation to inhibit β-glucan-induced IRF5 activation.** **a**, **b** Immunoblot and quantification analysis of the indicated protein phosphorylation (**a**) and nuclear IRF5 (**b**) in wild-type (*Clec2d*[+/+]) and *Clec2d*-deficient (*Clec2d*[−/−]) BMDCs stimulated with curdlan (20 μg/well) for indicated time. **c**, **d** Immunoblot and quantification analysis of the indicated protein phosphorylation (**c**) and nuclear IRF5 translocation (**d**) in RAW264.7 cells stably expressing mouse Clec2d or mock stimulated with curdlan (20 μg/well) for indicated time. **e** Upper panel: Immunoblot analysis of MyD88 degradation in wild-type and *Clec2d*-deficient BMDCs stimulated with curdlan (20 μg/well) for indicated time. Lower panel: Quantitative analysis of MyD88 normalized to GAPDH. **f** Upper panel: Immunoblot analysis of MyD88 degradation in RAW264.7 cells stably expressing mouse Clec2d or mock, which were pretreated with proteasome inhibitor (MG132, 20 μM) for 30 min and then stimulated with curdlan

(20 μg/well) for indicated time. Lower panel: Quantitative analysis of MyD88 normalized to GAPDH. **g** Upper panel: Immunoblot analysis of IRF5 phosphorylation in RAW264.7 cells o RAW264.7 cells stably expressing mouse Clec2d or mock, which were pretreated with MyD88 inhibitor (TJ-M2010-5, 40 μM) for 30 min and then stimulated with curdlan (20 μg/well) for indicated time. Lower panel: Quantitative analysis of p-IRF5 normalized to GAPDH. **h** Left panel: Immunoblot analysis of IRF5 phosphorylation and MyD88 degradation in wild-type, *Clec2d*-deficient, *TLR2*-deficient and *Clec2d/TLR2*-deficient BMDCs stimulated with curdlan (20 μg/well) for indicated time. Middle panel: Quantitative analysis of p-IRF5 normalized to GAPDH. Right panel: Quantitative analysis of MyD88 normalized to GAPDH. Data were presented as mean ± SEM; *n* = 3 (**a**–**h**) biologically independent samples. Data were analyzed by one-way ANOVA adjusted for multiple comparisons in (**a**–**h**). Source data are provided as a Source Data file.

We further investigated the role of IL-12 in the CLEC2D-mediated IFN-γ production and its impact on host defense against *C. albicans* infection. To achieve this, we employed a specific antibody to neutralize endogenous IL-12. Our findings showed that neutralizing endogenous IL-12 had a profound effect on *Clec2d*-deficient mice, leading to a significant increase in fungal burdens in the kidneys and brains upon *C. albicans* infection (Fig. 6g). However, no significant

changes were observed in wild-type mice under similar conditions (Fig. 6g). Furthermore, the neutralization of IL-12 resulted in a significant reduction in IFN-γ production in the kidneys of *Clec2d*-deficient mice, while no alterations were observed in the wild-type mice (Fig. 6h). These results strongly indicated the critical role of IL-12 in the CLEC2D-mediated IFN-γ production and subsequent host defense against *C. albicans* infection.

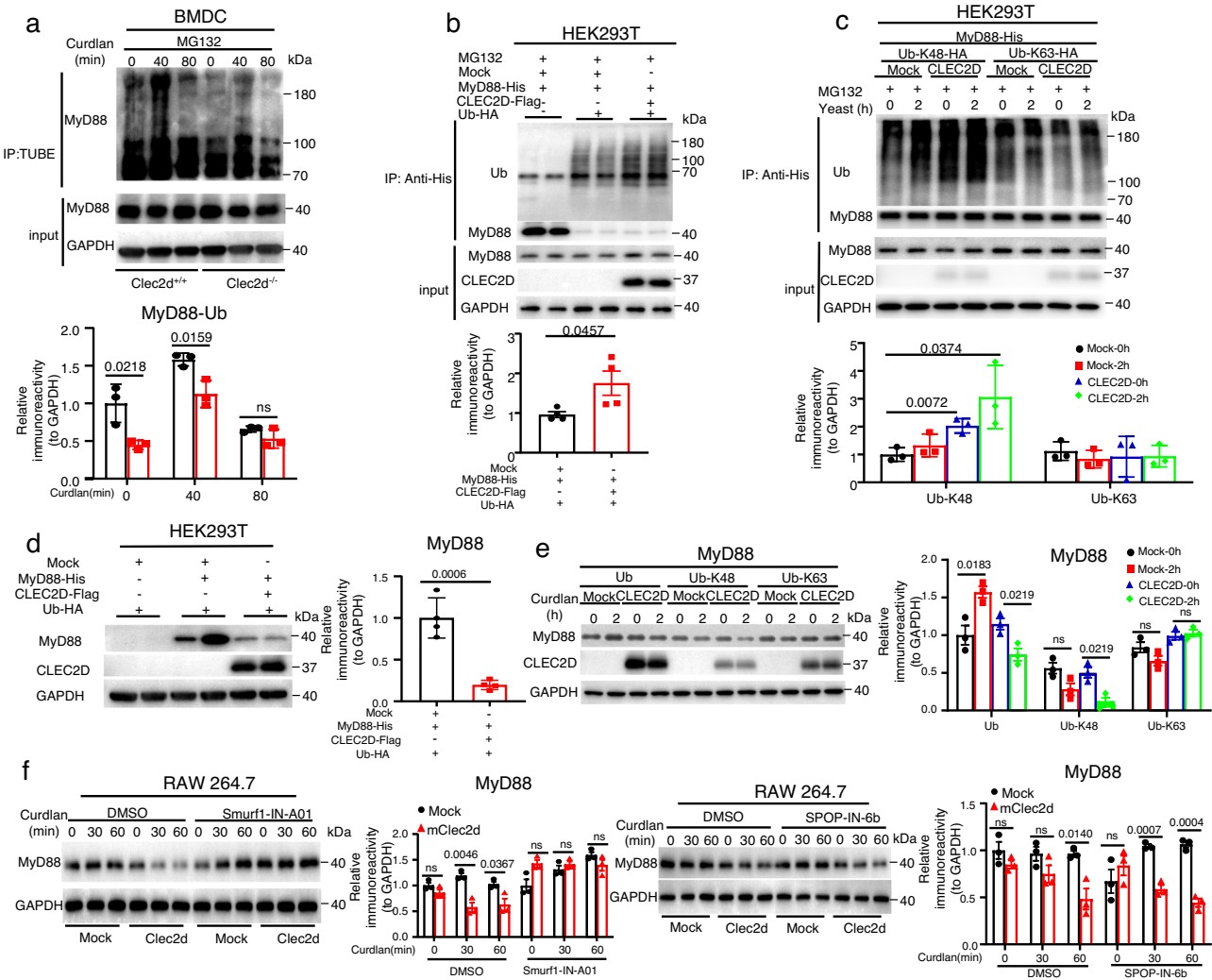

**Fig. 4 | CLEC2D promotes Smurf1-mediated degradation of MyD88 through K48-linked ubiquitination. a** Immunoblot analysis of curdlan (20 μg/well)-induced ubiquitination of MyD88 in wild-type and *Clec2d*-deficient BMDCs with magnetic beads-coupled Tandem Ubiquitin Binding Entities (TUBEs). Three times experiments were repeated independently with similar results. **b, c** Immunoblot analysis of the degradation of MyD88 expressed in HEK293T cells with co-expression of MyD88-his, Clec2d-Flag and Ub-HA (**b**) or Ub-K48-HA and Ub-K63-HA (**c**) in the presence of MG132 (20 μM). Three times experiments were repeated independently with similar results. **d–e** Immunoblotting and quantification analysis of the degradation of MyD88 expressed in HEK293T cells with co-expression of

MyD88-His, Clec2d-Flag and Ub-HA (**d**) or Ub-K48-HA and Ub-K63-HA (**e**). **f** Immunoblotting and quantification analysis of the degradation of MyD88 in RAW264.7 cells stably expressing mouse Clec2d or mock, which were pretreated with Smurf1 inhibitor (Smurf1-IN-A01, 10 μM), SPOP inhibitor (SPOP-IN-6b dihydrochloride, 10 μM) for 30 min and then stimulated with Curdlan (20 μg/well) for indicated time. Data were presented as mean ± SEM; *n* = 3 (**e, f**), *n* = 4 (**d**) biologically independent samples. Data were analyzed by one-way ANOVA adjusted for multiple comparisons in (**e, f**), Data were analyzed by unpaired two-sided Student's *t*-test in (**d**). Source data are provided as a Source Data file.

In order to investigate the role of NK cells in *Clec2d*-deficient mice during *C. albicans* infection, we conducted a successful depletion of NK cells using an anti-NK1.1 antibody. We found that the depletion of NK cells in both wild-type and *Clec2d*-deficient mice resulted in a higher fungal burden in the kidney and brain compared to mice receiving control IgG after being infected with *C. albicans* (Fig. 6i). These results provided further confirmation that CLEC2D negatively regulated the production of IL-12, consequently impairing the generation of IFN-γ-producing NK cells. This ultimately led to the suppression of the host's antifungal immune response against *C. albicans* infection.

**Lyz2⁺ myeloid cell-specific deficiency of *Clec2d* promoted survival of *C. albicans*-infected mice**
To further investigate the contribution of myeloid CLEC2D in the immune response against *C. albicans* infection, we utilized a mouse model with specific deficiency of CLEC2D in myeloid cells

(*Clec2d*^fl/fl^*Lyz2*^Cre/+^). This was achieved by generating Clec2d^fl/fl^ mice and crossing them with Lyz2^Cre/+^ mice (Supplementary Fig. 6). We observed that *Clec2d*^fl/fl^*Lyz2*^Cre/+^ mice exhibited significant increases in survival and lower fungal burden in the kidneys compared to control mice (Clec2d^fl/fl^) after *C. albicans* infection (Fig. 7a, b). Histological analysis further confirmed that myeloid cell-specific deficiency of *Clec2d* led to the reduced renal inflammation and fungal cell presence (Fig. 7c). Importantly, the deficiency of Clec2d in myeloid cells resulted in a significant increase in the levels of IL-12 and IFN-γ, but not IL-12p40, along with augmented numbers and frequency of NK cells and IFN-γ-producing NK cells on Day 1 following *C. albicans* infection (Fig. 7d–f).

Conversely, treatment with IRF5 inhibitor N5-1 significantly increased fungal burden in the kidneys and brains of *Clec2d*^fl/fl^*Lyz2*^Cre/+^ mice, accompanied by decreased levels of IL-12 and IFN-γ in the kidneys of *Clec2d*^fl/fl^*Lyz2*^Cre/+^mice on Day 1 after infection with *C. albicans* (Fig. 7g, h). However, this treatment had no effect on the fungal burden

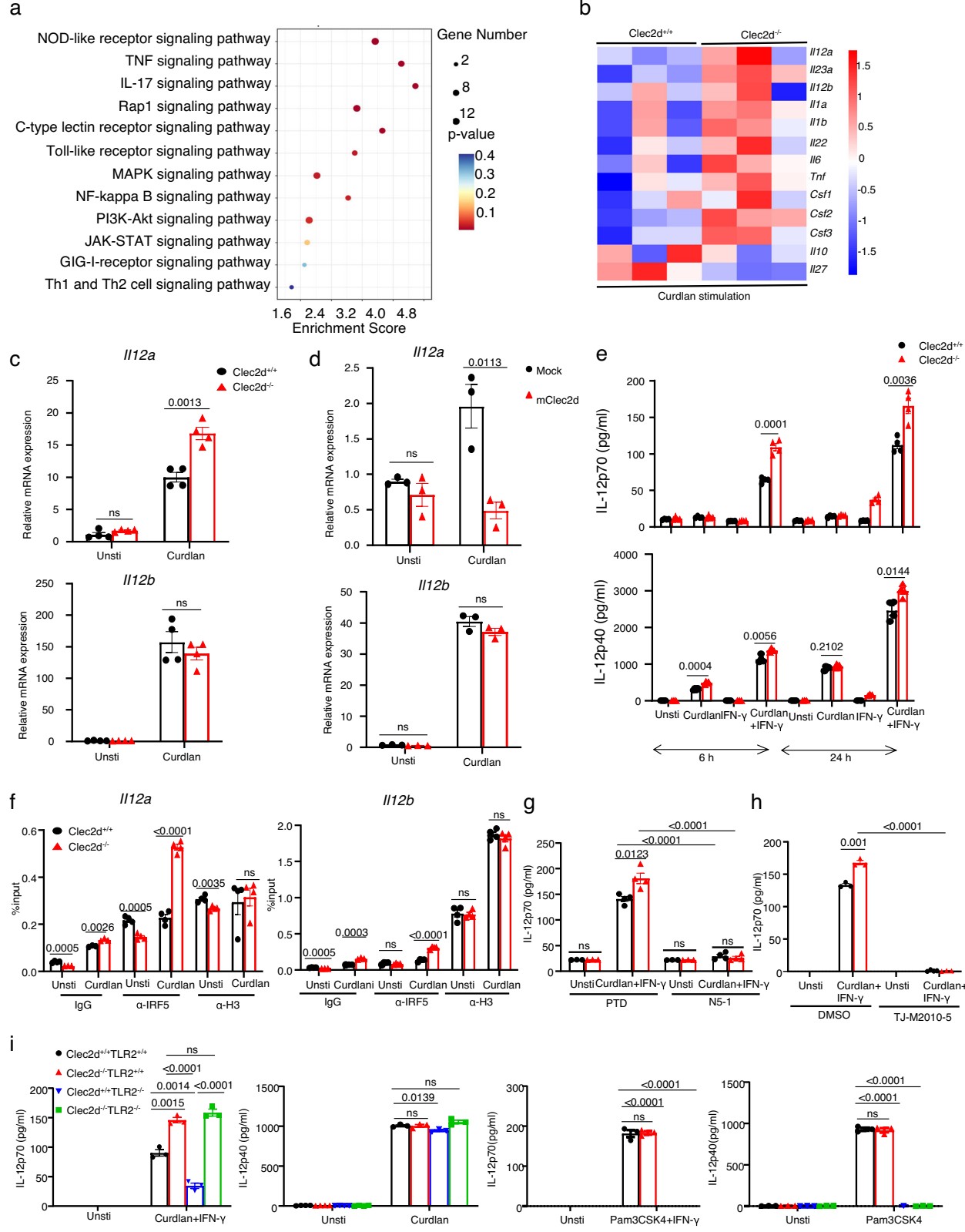

in the kidneys and brains of control *Clec2d*<sup>fl/fl</sup> mice, and instead, it significantly increased the levels of IL-12 and IFN-γ in the kidneys after infection with *C. albicans* (Fig. 7g, h). Together, these data confirmed that CLEC2D expressed on myeloid cells negatively regulated IRF5-mediated IL-12 production and subsequently limited the generation of IFN-γ-producing NK cells, thereby suppressing host defense against *C. albicans* infection.

## Discussion

The study of innate recognition offers much promise for understanding and manipulating immunity. Based on their intracellular signaling motifs, myeloid CLRs can be classified into the following broad categories: immunoreceptor tyrosine-based activating motif (ITAM)-coupled CLRs, hemi-ITAM-bearing CLRs, immunoreceptor tyrosine-based inhibitory motif (ITIM)-containing CLRs, and a group of CLRs

**Fig. 5 | CLEC2D/TLR2 heterodimers and CLEC2D homodimers negatively regulate IRF5-mediated IL-12 production. a** KEGG analysis of differentially expressed genes involved in signaling transduction in wild-type or *Clec2d*-deficient BMDCs stimulated with curdlan (20μg/well) for 3 h. **b** Heatmaps show of differentially expressed genes encoding proinflammatory cytokines in wild-type or Clec2d-deficient BMDCs stimulated with curdlan (20μg/well) for 3 h. **c** Real-time quantitative PCR analysis of *Il12a* and *Il12b* mRNA expression in wild-type and *Clec2d*-deficient BMDCs stimulated with curdlan (20 μg/well) for 3 h. **d** Real-time quantitative PCR analysis of *Il12a* and *Il12b* mRNA expression in RAW264.7 cells stably expressing mouse Clec2d or mock stimulated with curdlan (20 μg/well) for 3 h. **e** ELISA results of IL12p70 and IL-12p40 in the supernatants of wild-type and *Clec2d*-deficient BMDCs, which were stimulated with curdlan (5 μg/well), IFN-γ (20 ng/well), curdlan plus IFN-γ for 6 or 24 h. **f** Chromatin immunoprecipitation (with control IgG, anti-IRF5 or anti-H3 as positive control; horizontal axis) and PCR analysis of the binding of IRF5 to the promoters of *il12a* and *il12b* in wild-type and *Clec2d*-deficient BMDCs stimulated with curdlan (20 μg/well) for 3 h. **g, h** ELISA of IL12p70 in the supernatants of wild-type and *Clec2d*-deficient BMDCs, which were pretreated with IRF5 inhibitory peptide (N5-1, 50 μM) and control peptide (PTD, 50 μM) (**g**) or MyD88 inhibitor (TJ-M2010-5, 40 μM) (**h**) and then stimulated with curdlan plus IFN-γ for 6 h. **i** ELISA of IL-12p70 and IL-12p40 in the supernatants of wild-type, *Clec2d*-deficient, *TLR2*-deficient and *Clec2d/TLR2*-deficient BMDCs stimulated with curdlan or Pam3CSK4 (100 ng/ml) plus IFN-γ (20 ng/well) for 6 h. ns, no significance. Data were presented as mean ± SEM; *n* = 3 (**a, b, d, g** Unsti group, **h, i**), n = 4 (**c, e, f, g** Curdlan + IFN-γ group) biologically independent samples. Data were analyzed by one-way ANOVA adjusted for multiple comparisons in (**a–i**). Source data are provided as a Source Data file.

lacking typical signaling motifs. It has been reported that CLEC2D lacks ITAM/ITIM motifs or known signaling adaptors used by other stimulatory CLRs. Here, we identified CLEC2D as a specific receptor for β-glucans and α-mannans from *C. albicans*. Moreover, we show that CLEC2D can form homodimers independently, as well as heterodimers with TLR2. Of note, homo- or hetero-dimeric CLEC2D has high binding ability to β-glucans and low binding ability to α-mannans. We further identified CLEC2D as a suppressor of proinflammatory cytokine IL-12. At the molecular level, we identified CLEC2D as a suppressor of IFR5 activation induced by β-glucans, but not α-mannans, and IRF5 as the crucial factor for binding the *Il12a* promoter. Thus, myeloid cell-specific deficiency of *Clec2d* results in a significant survival advantage of mice after bloodstream infection with *C. albicans* due to the increased IL-12 production and subsequent generation of IFN-γ-producing NK cells. Together, we conclude that CLEC2D functions as an inhibitory CLR lacking ITIM motifs, either through homodimers or heterodimers with TLR2, to negatively regulate antifungal immunity through inhibiting IRF5-mediated IL-12 production. However, it remains lack of in vivo experimental means to distinguish the contribution of CLEC2D/TLR2 homo- or hetero-dimers to the negative regulation of antifungal immunity.

Dectin-1 contains a hemi-ITAM motif in its intracellular tail. Signaling through Dectin-1, which is largely mediated through Syk kinase, is thought to be sufficient for production of cytokines, such as IL-10, IL-6 and IL-23[33,34], whereas others, such as TNFα, additionally require the recognition of another undefined fungal component by TLR2, and signaling through the MyD88 pathway[35]. However, it remains elusive about the mechanisms underlying the regulation of IL-12 production during fungal infections. In macrophages and DCs, Dectin-1 and TLR2 are synergistic in mediating the production of cytokines such as IL-12 and TNFα[16]. Paradoxically, co-stimulation of Dectin-1 and TLR2 can suppress the production of IL-12 in BMDCs[36,37]. A previous study has also shown that Syk-deficient BMDCs can potently produce IL-12 upon *C. albicans* yeast stimulation, indicating that the Dectin-1/Syk and Dectin-1/TLR2 pathways can operate independently[38]. Here, we show that CLEC2D interacts with itself or TLR2 to form homo- or hetero-dimers, respectively. As expected, either CLEC2D or TLR2 has no interaction with Dectin-1. Importantly, the deficiency of either *Clec2d* alone or in combination with *TLR2* deficiency in BMDCs resulted in a significant increase in IL-12 production upon β-glucan stimulation, whereas TLR2 deficiency alone in BMDCs significantly impaired IL-12 production. Thus, we predict that CLEC2D forms homodimers or heterodimers with TLR2 to negatively regulate IL-12 production during fungal infections. However, it needs further investigation whether Dectin-1 is involved in the negative regulation of IL-12 production mediated by CLEC2D/TLR2 homo- or hetero-dimers.

It has been well-documented that IRF5 directly interacts with and is activated by MyD88 and functions as a key transcription factor for TLR-dependent induction of proinflammatory cytokine genes, such as IL-12[39]. Two recent studies show that Dectin-1 activation by *C. albicans* triggered IRF5-mediated expression of IFN-β in DCs[40,41]. Our present study shows that *Clec2d* deficiency results in a significant increase of β-glucan-induced IL-12 production by IFN-γ primed BMDCs through enhancing the binding of IRF5 to *Il12a* promoters. Importantly, inhibition of either IRF5 or Myd88 activation with their respective inhibitors N5-1[31] or TJ-M2010-5[26] completely blocks β-glucan-induced IL-12 production in IFN-γ primed wild-type and *Clec2d*-dificient BDMCs. Thus, these findings suggest that CLEC2D negatively regulates IRF5-mediated IL-12 production induced by β-glucans.

PRRs form homo- and heterodimers to exert the potential for modulating ligand bindings. Among the TLR family, TLR2 has been well-characterized to heterodimerize with either TLR1 or TLR6 for efficient ligand binding and discrimination of triacyl and diacyl lipopeptides from bacteria[3,4]. Apart from that, in the CLR family, our previous study has shown that homo- and heterodimerization of Dectin-2 and Dectin-3 respectively display their low and high binding ability to α-mannans of *C. albicans*[9]. In our present study, we show that CLEC2D interacts directly with itself or TLR2 to form homodimers, as well as heterodimers with TLR2. More excitingly, both CLEC2D homodimers and CLEC2D/TLR2 heterodimers have a higher binding ability to β-glucans compared to TLR2 homodimers. However, these dimers exhibit similarly low binding ability to α-mannans. Furthermore, CLEC2D enhances the binding ability of heterodimeric TLR2 to β-glucans compared to TLR2 homodimers. These data reveal a new example of PRR dimerization cross family to regulate ligand bindings. However, further studies are needed to unravel the structural basis for ligand recognition by CLEC2D/TLR2 heterodimers and CLEC2D-homodimers.

Ligand-dependent dimerization of TLRs is an initial step in immune modulation of their downstream signaling, and MyD88 is the most universal adaptor molecule in TLR signaling. However, the association of MyD88 with TLR2 alone has been reported to be barely detectable, and Pam3CSK4 stimulation greatly enhanced the recruitment of MyD88 to TLR2/TLR1 heterodimers[42]. Our present study shows that stimulation of homo- and heterodimeric CLEC2D with β-glucans promotes proteasomal degradation of Myd88 through facilitating its K48-linked, but not K63-linked ubiquitination. In addition, *Clec2d* deficiency in BMDCs selectively leads to the increased phosphorylation and nuclear translocation of IRF5 induced by β-glucans, but not α-mannans or Pam3CSK4. Moreover, the deficiency of either *Clec2d* alone or in combination with *TLR2* in BMDCs enhances β-glucan-induced IRF5 phosphorylation and MyD88 stabilization, whereas *TLR2* deficiency alone has no influence on IRF5 activation and MyD88 degradation. Thus, we conclude that CLEC2D forms homo- or heterodimers with TLR2 to mediate the degradation of MyD88 and subsequently to inhibit IRF5 activation induced by β-glucans.

It has been reported that the activation of MAPK, Syk and NF-κB signaling in macrophages is independent of CLEC2D upon stimulation with purified histones[21]. Our recent study has shown that CLEC2D mediates the phosphorylation of p38 in MDSCs when stimulated with glucuronoxylomannan of *Cryptococcus neoformans*[23]. However, our current study shows that *Clec2d* deficiency in BMDCs slightly increases

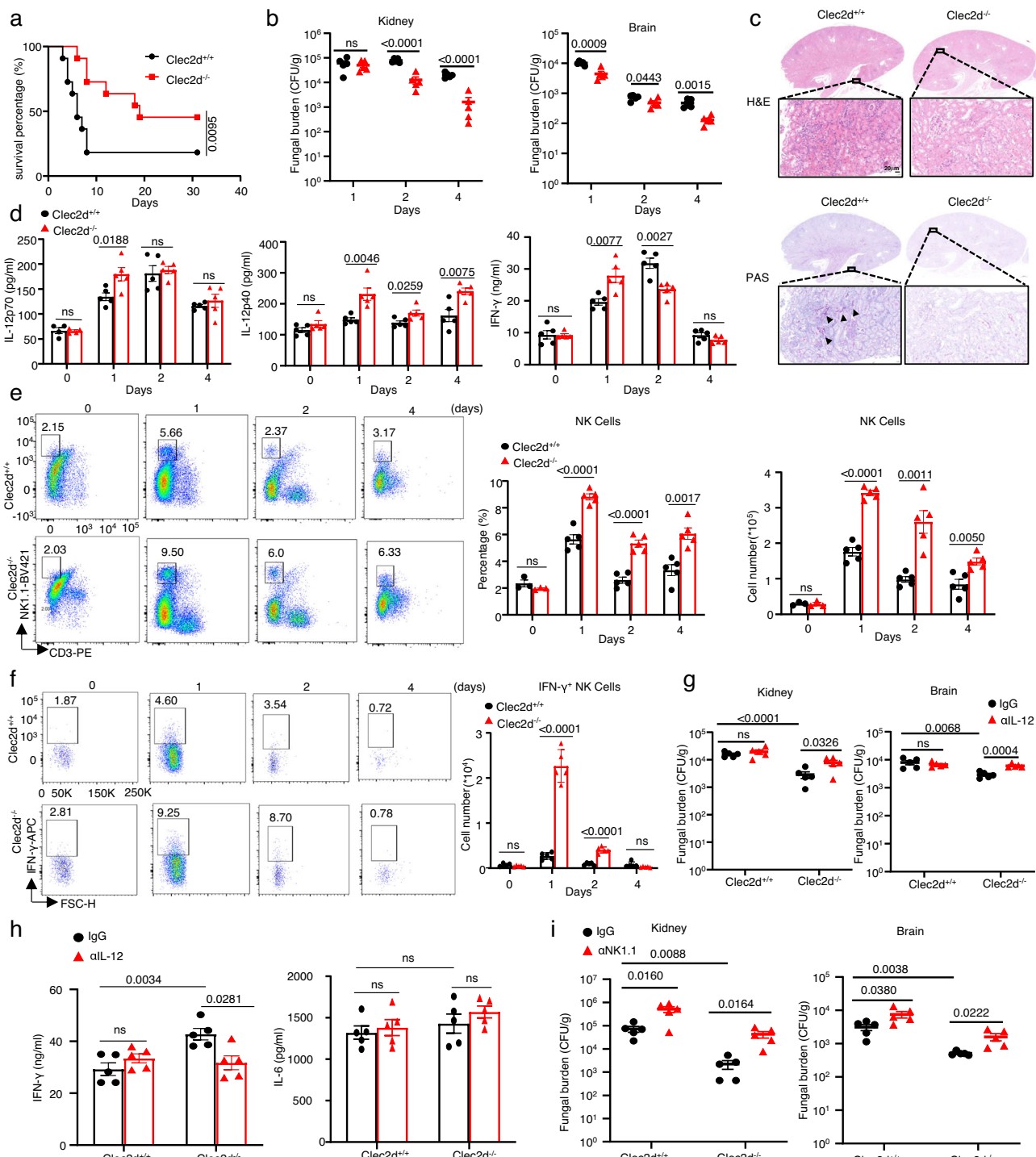

**Fig. 6 | *Clec2d* knockout mice are highly resistant to infection with *C. albicans* through increasing IL-12 production. a–c** Survival (**a**), fungal burden in kidney and brain on Day 2 (**b**), representative histological images with hematoxylin-eosin (H&E) and Periodic Acid-Schiff (PAS) staining of kidney at Day 2 (**c**) of wild-type and *Clec2d*-deficient mice intravenously infected with *C. albicans* (SC5314, 2 × 10⁵ CFU). Scar bars = 20 μm. Arrows indicate fungi hyphae. **d–f** ELISA of IL-12p70, IFN-γ, IL-12p40 (**d**), flow assay of frequency and numbers of NK cells (**e**) and numbers of IFN-γ⁺ NK cells (**f**) in kidneys of wild-type and *Clec2d*-deficient mice after infection with *C. albicans* for indicated time. **g, h** Fungal burden in kidneys on Day 2 and brains on Day 1 (**g**), ELISA of IFN-γ and IL-6 in kidneys on Day 1 (**h**) of *C. albicans*-infected wild-type and *Clec2d*-deficient mice receiving IL-12 neutralizing antibody (αIL-12, 250μg/mouse) or control IgG. **i** Fungal burden in kidneys and brains on Day 2 of *C. albicans*-infected wild-type and *Clec2d*-deficient mice administrated with NK1.1 neutralizing antibody (αNK1.1350 μg/mouse) or control IgG. ns, no significance. Data were presented as mean ± SEM; *n* = 11(**a**), *n* = 3 (**e** day0 group), *n* = 4 (**d** IL-12p70 on Day 0 group), *n* = 5 (**b**, **d–i**) biologically independent samples. Data were analyzed by one-way ANOVA adjusted for multiple comparisons in (**b**), (**d–i**), Log-rank Mantel-Cox test in (**a**). Source data are provided as a Source Data file.

β-glucan-induced phosphorylation of p38, JNK and ERK1/2, which is not affected by the overexpression of Clec2d in RAW264.7 cells. Previous studies have identified E3 ligases, including Smurf1 and SPOP, that modulate TLR signaling by promoting the polyubiquitination and

degradation of MyD88 through the proteasome pathway[27,43]. Our present study shows that inhibiting Smurf1, but not SPOP, completely blocks Clec2d-mediated degradation of Myd88 induced by β-glucans, suggesting that Clec2d promotes MyD88 degradation through

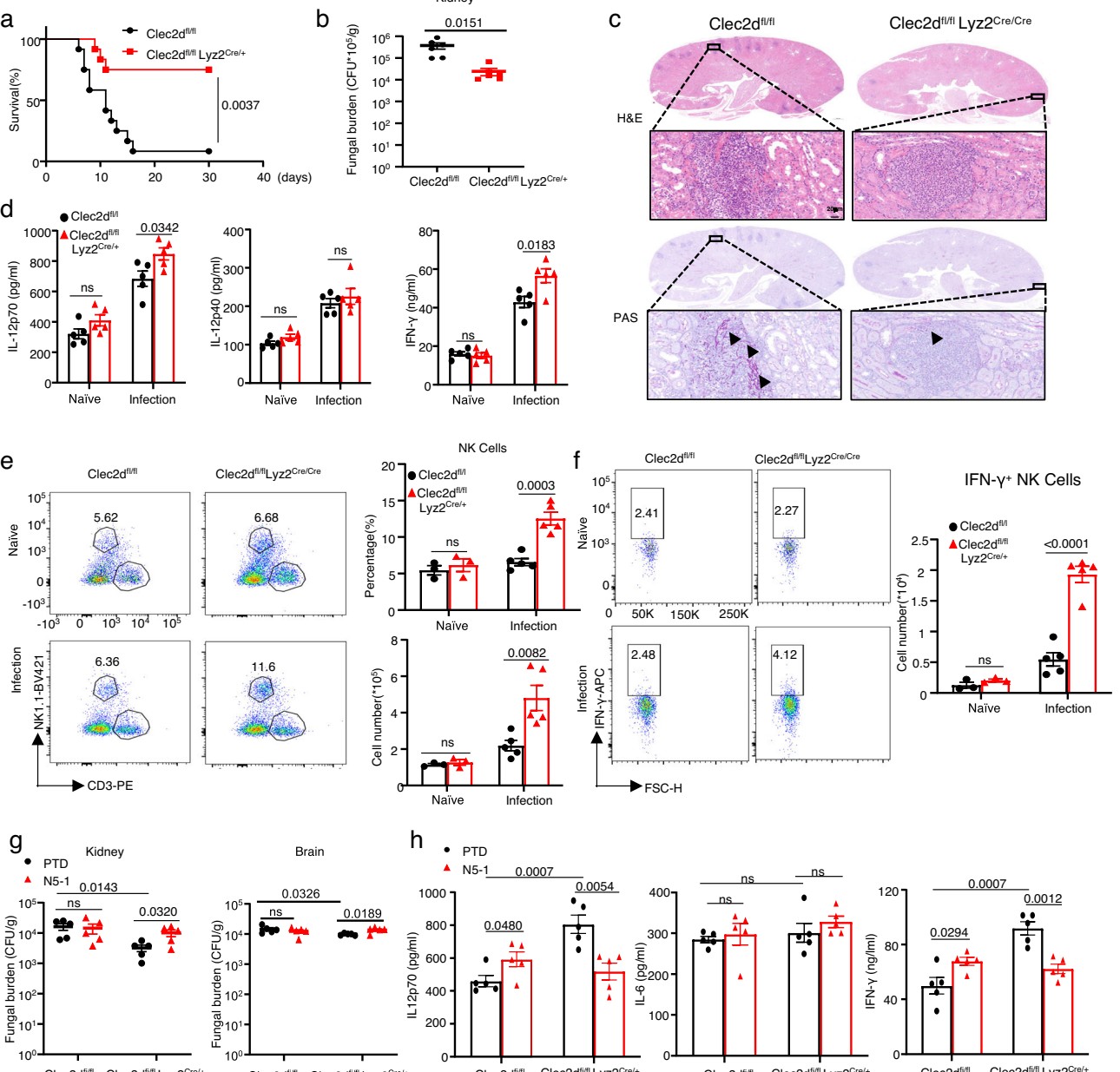

**Fig. 7 | Lyz2⁺ myeloid cell-specific deficiency of Clec2d promoted survival of *C. albicans*-infected mice. a, b** Survival (**a**) and fungal burden in kidneys on Day 2 (**b**) of *Clec2d*^fl/fl *Lyz2*^Cre/+ and *Clec2d*^fl/fl mice, which were intravenously infected with *C. albicans* (SC5314, 2 × 10⁵ CFU). **c–f** Representative histological images with hematoxylin-eosin (H&E) and Periodic Acid-Schiff (PAS) staining on Day 2 (**c**), ELISA of IL-12p70, IL-12p40, IFN-γ in kidney tissue homogenate on Day 1 (**d**), flow cytometry assay of the frequency and numbers of NK cells (**e**) and the numbers of IFN-γ⁺ NK cells (**f**) of kidneys on Day 1 in *Clec2d*^fl/fl *Lyz2*^Cre/+ and *Clec2d*^fl/fl mice, which were intravenously infected with *C. albicans*. Scar bars = 20 µm. Arrows indicate fungi

hyphae. **g, h** Fungal burden of kidneys on Day 2 and brains on Day 1 (**g**) or ELISA of IL-12p70, IFN-γ, IL-6 in kidney homogenate on Day 1 (**h**) of *Clec2d*^fl/fl *Lyz2*^Cre/+ and *Clec2d*^fl/fl mice, which were pretreated with IRF5 inhibitory peptide N5-1 (100 µg/mouse) or control PTD and then intravenously infected with *C. albicans* (2 × 10⁵ CFU). ns, no significance. Data were presented as mean ± SEM; *n* = 12 (**a**), *n* = 3 (**e** and **f** Naive group), *n* = 5 (**b, d, e** and **f** infected group, **g, h**) biologically independent samples. Data were analyzed by Log-rank Mantel-Cox test in (**a**), unpaired two-sided Student's *t*-test in (**b**), one-way ANOVA adjusted for multiple comparisons in (**d–h**). Source data are provided as a Source Data file.

recruiting Smurf1. A previous study shows that Smad6 mediates TGF-β1-induced K48-linked polyubiquitination and degradation of Myd88 through recruiting Smurf1[27]. It has been well-documented that Smad6 acts as an important mediator to negatively regulate TLR or IL-1R signaling[27,44]. Future studies will focus on confirming the hypothesis that CLEC2D may form a complex with Smad6 to recruit Smurf1 for mediating β-glucan-induced K48-linked polyubiquitination and degradation of Myd88.

*C. albicans* is an opportunistic human fungal pathogen when the immune system of the host is compromised. Invasive candidiasis is the

fourth most common cause of bloodstream in nosocomial infections. The most well-studied CLR against *C. albicans* infections is Dectin-1, which is originally described as the β-glucan receptor expressed in myeloid cells and plays a crucial role in eliciting antifungal proinflammatory immune responses[33,45]. Moreover, it has been shown that Galectin-3 negatively regulates Dectin-1-mediated IL-23 production in DCs to suppress Th17 polarization against *C. albicans* infection and CLEC-1 negatively regulates Dectin-1-mediated IL-1β production and neutrophils recruitment in response to *Aspergillus fumigatus* infection[46,47]. In this study, we show that CLEC2D forms homodimers or

heterodimers with TLR2 to specifically recognize β-glucans exposed on the cell surface of *C. albicans* yeast. Notably, myeloid cell-specific deficiency of *Clec2d* results in a significant survival advantage of mice after bloodstream infection with *C. albicans* due to the increased IRF5-mediated IL-12 production and subsequent generation of IFN-γ-producing NK cells. Moreover, neutralization of endogenous IL-12 using its specific antibody or blockade of IRF5 activation with its inhibitor N5-1 significantly increases the susceptibility of *Clec2d*-knockout, but not wild-type, mice to bloodstream infection with *C. albicans* accompanied by the decreased amount of IFN-γ production in the kidneys. However, there are conflicting reports regarding the susceptibility of *TLR2* knockout mice to disseminated candidiasis. In one study, *TLR2* knockout mice have been shown to be more resistant to disseminated *C. albicans* infection due to the increased chemotaxis and enhanced candidacidal capacity of macrophages[48]. Other authors have shown that *TLR2* knockout mice are highly susceptible to intraperitoneal or intravenous infection with *C. albicans* due to the impaired production of TNF-α and macrophage inhibitory protein-2 (MIP-2) by macrophages[49]. Here, we conclude that CLEC2D forms homodimers or heterodimers with TLR2 to negatively regulate IRF5-mediated antifungal immunity against *C. albicans* infection.

Traditionally, TLRs and CLRs are considered as stimulatory receptors that upregulate the production of proinflammatory cytokines. Our present data suggest that CLEC2D forms homodimers or heterodimers with TLR2 to function as inhibitory receptors to suppress IRF5-mediated IL-12 production, thereby negatively regulating

antifungal immunity against *C. albicans* infection (Fig. 8). Understanding the functional diversity of PRR dimerization and activation may be useful in the development of new immunotherapeutic strategies to combat these devastating infections.

## Methods

### Ethics statement

All animal experiments were performed according to the protocol approved by the Animal Ethics Committee of Tongji University School of Medicine (protocol No. TJAA09023101). Studies of human PBMCs were approved by the Human Research Committee of Tongji University School of Medicine (protocol 2023TJDXSY017).

### Mice

Female C57BL/6J mice with 6–8 weeks old were procured from Shanghai SLAC animal laboratory. *Clec2d*[−/−] mice line was generated using CRISPR−Cas9 methods, as previously described[23]. *Lyz2*[Cre] and *Clec2d*[fl/fl] mice were purchased from GemPharmatech (Nanjing, China), *Clec2d*[fl/fl] mice were crossed to *Lyz2*[Cre] to obtain *Lyz2*[Cre/+]*Clec2d*[fl/fl]. *TLR2* knockout mice were purchased from The Jackson Laboratory (USA). To obtain *Clec2d*[−/−]*TLR2*[−/−] mice, *Clec2d*[−/−] mice were crossbred with *TLR2*[−/−] mice. All knockout mice used in this study were derived from a C57BL/6J background and were genotype confirmed by PCR analysis using DNA extracted from tail tissue. All mice used in this study were bred and housed in animal facilities located at Tongji University School of Medicine in Shanghai, China. The animal housing

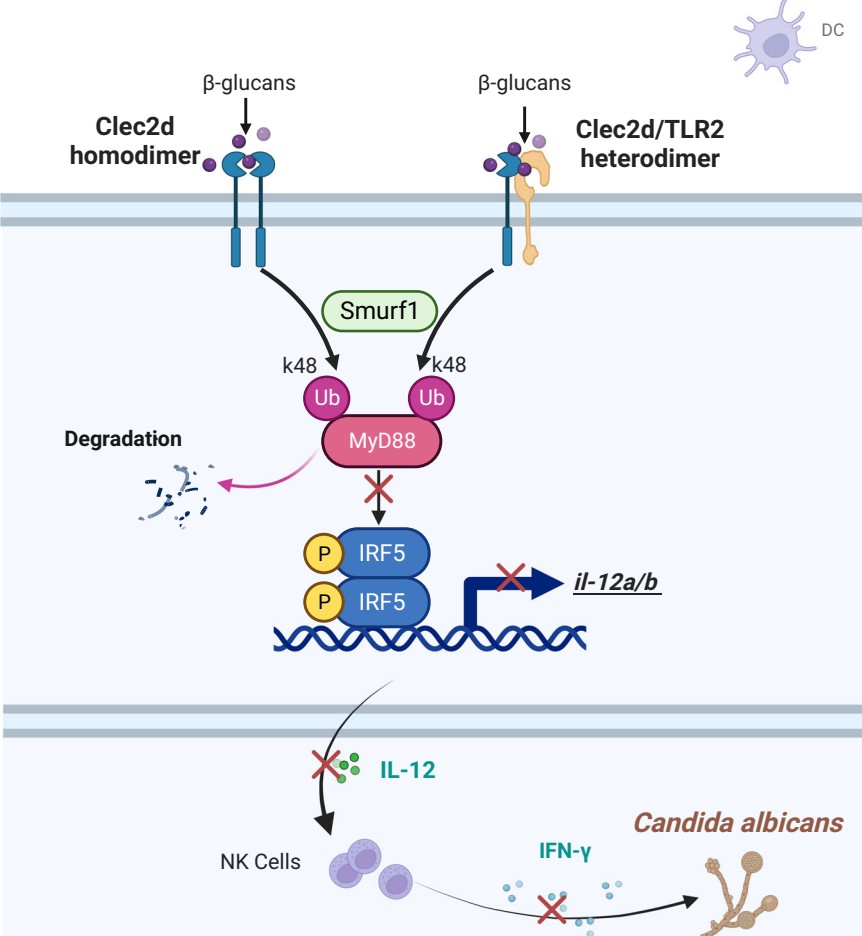

**Fig. 8 | Schematic diagram of the mechanism.** During *C. albicans* infection, CLEC2D in dendritic cells forms homodimers or heterodimers with TLR2 to recognize β-glucan derived from *C. albicans*. This process triggers Smurf1-

mediated K-48 linked ubiquitin degradation of MyD88, thereby inhibiting the activation of transcription factor IRF5 and subsequent IL-12 production. Created with BioRender.com.

environment was maintained under specific pathogen-free conditions with a 12-h light and 12-h dark cycle. All mice were provided with free access to food and water to ensure adequate nutrition and hydration. All animal studies were performed using sex-and age-matched female mice.

## Antibodies, plasmids, reagents

The cDNAs of human TLR2, human MyD88, human CLEC2D, and human Dectin-1 were synthesized by GENEWIZ in Suzhou, China. Human TLR2 was successfully cloned into pEF4 C-mychisB, while human Dectin-1 was cloned into pCMV-HA. Human MyD88 and human CLEC2D were both cloned into either pLV-IRES-EGFP or pcDNA6 cs2.0 C-Flag. All primers used for amplifying genes are listed in Supplemental Table 1.

Curdlan (C7821, Lot#039M4117V) and α-mannan (M7504, Lot#SLBT8710) were both procured from Sigma-Aldrich. Pam3CSK4 (tlrl-pms) was obtained from invivogen, and the recombinant murine granulocyte-macrophage colony-stimulating factor (GM-CSF) was purchased from PeproTech.

## Fungal strains and culture conditions

The standard strain *C. albicans* SC5314 was procured from ATCC (American Type Culture Collection). The *C. albicans* yeast cells were cultivated in YPD medium at 30 °C for 16-18 h. After that, *C. albicans* cells were washed with PBS three times and counted using a hematocytometer. To obtain heat-killed *C. albicans*, yeast cells were incubated at 95 °C for one hour. For hyphae, yeast cells were cultured in RPMI 1640 medium for three hours.

## Cell lines

HEK293T and Raw264.7 cells(Jia's lab) were cultivated in 10 cm petri dish with DMEM supplemented with 10% FBS, 100 U/mL penicillin, and 100 μg/mL streptomycin, and maintained at 37 °C with 5% $CO_2$ in a humidified environment. When the cells grew to cover 80−90% of the area of the dish, HEK293T cells were treated with 2 mL of 0.25% trypsin, and Raw264.7 cells were digested using 2 mL of TEN buffer (150 mM NaCl, 40 mM Tris, 5 mM EDTA). Subsequently, half of the cells were passaged into a new petri dish.

## Mouse model of systemic candidiasis

Mice were infected with $3 \times 10^5$ CFU of *C. albicans* through the lateral tail vein and monitored daily for health and survival. For fungal burden determination, mice were infected with $2 \times 10^5$ CFU of *C. albicans*, and their fungal burden was analyzed. In addition, in vivo cytokine measurements, flow cytometry assays, and histopathological analysis were carried out.

## Fungal burden determination

The kidneys were aseptically removed, weighed, and then homogenized in 1 ml of PBS using a high-speed cryogenic tissue grinding machine (servicebio, China). The fungal burden was determined by plating organ homogenates in serial dilutions on Sabouraud dextrose agar (SDA) plates. The plates were then incubated for 24 h at 37 °C, and the colony forming units (CFUs) were counted.

## Histology

The kidneys were fixed with 4% paraformaldehyde and then embedded in paraffin. All staining procedures and image scanning were performed by Servicebio. PAS and H&E images were analyzed using Caseviewer.

## In vivo treatment with anti-IL-12 and anti-NK1.1 antibody

Mice were given a pre-injection of 250 μg of anti-IL-12 (BioXcell, BE0233) and 350 μg of anti-NK1.1 (BioXcell, BE0036) neutralizing antibody, or an isotype control antibody (BioXcell, BE0090, BE0085) 12 h prior to being infected with *C. albicans*.

## In vivo treatment with IRF5 peptide inhibitor N5-1

Mice were pre-injected intraperitoneally with 100 μg of IRF5 peptide inhibitor (DRQIKIWFQNRRMKWKKPRRVRLK, synthesized by GL Biochem, Shanghai, China) for two consecutive days as per the previously reported strategies[31]. Alternatively, control peptide PTD (DRQI-KIWFQNRRMKWKK) was used. After infection with *C. albicans*, mice were daily injected intraperitoneally with an IRF5 peptide to monitor their health and survival.

## Preparation of kidney cell suspension

To analyze renal leukocyte infiltrates, cell suspensions were obtained from organ homogenates that were digested with 2 mg/ml of VI type Collagenase (Roche, 11088866001) for 90 min at 37 °C. The resulting mixture was filtered through 40-μm cell strainers and washed twice in 2% FBS-containing PBS. The cells were then resuspended in 40% percoll (Cytiva, 17089109) and centrifuged on 80% percoll for 20 min at 1500 g. The supernatant tissue suspension was removed, and the cells were washed with 2% FBS-containing PBS. Any remaining red blood cells were removed by treatment with RBC (red blood cell lysate).

## Flow cytometry assay

To detect IFN-γ production by NK cells, cells were stimulated for 4 h with 5 μg/ml Brefeldin A (Yeasen, 50502ES03), then stained with FITC-conjugated anti-CD45 (Biolegend, 147710, dilution 1:200), PE-conjugated anti-CD3 (Biolegend, 100205, dilution 1:200), and BV421-conjugated anti-NK1.1 (Biolegend, 108741, dilution 1:200) antibodies. The cells were then fixed with 4% paraformaldehyde and permeabilized using permeabilization buffer (Invitrogen, 88-8824-00). Finally, the cells were stained with APC-conjugated anti-IFN-γ antibody (Biolegend, 505810, dilution 1:100). Data were acquired on a cytometer (FACSCelesta, BD Biosciences) and analyzed using FlowJo X software (Tree Star, Ashland, OR).

## Preparation of BMDCs

To obtain bone marrow-derived dendritic cells (BMDCs), bone marrow cells from the femur and tibia were cultured in RPMI 1640 medium with 10% FBS and 20 ng/mL GM-CSF. The cells were plated in 12-well plates at a density of $0.5 \times 10^6$ cells and half of the medium was refreshed with GM-CSF-containing complete RPMI 1640 medium on Day 3 and Day 6. On Day 8, non-adherent cells were harvested for further analysis. The resulting BMDCs were characterized as CD11c$^+$ cells with a purity of over 80%.

## Stimulation of BMDCs and Raw264.7 cells

In order to perform ELISA measurement, 5 μg of curdlan was dissolved in absolute ethyl alcohol and coated onto 48-well plates. After coating, $3 \times 10^5$ BMDCs or Raw264.7 cells were stimulated for 6 h.

For Western blot assays, 20 μg of curdlan was coated onto 12-well plates. After coating, $3 \times 10^6$ BMDCs or Raw264.7 cells were seeded in the wells and stimulated with curdlan for the indicated time.

For co-immunoprecipitation, 200 μg of curdlan was coated onto a 10 cm petri dish. After coating, $5 \times 10^7$ BMDCs or Raw264.7 cells were stimulated with curdlan for the indicated time.

## In vitro ELISA binding assay

ELISA plates were coated with Curdlan or α-mannan at a concentration of 5 μg per well and incubated overnight at 4 °C. The plates were then blocked with 5% nonfat milk at room temperature for 1 h. CLEC2D-Fc, TLR2-His or Dectin-1-Fc proteins were added to the wells and incubated for 2 h. After washing, either anti-human Fc antibody or anti-His antibody was used as detection antibodies, with dilutions of 1:1000 for both antibodies. The purpose of this experiment was to assess the binding affinity of these proteins to the coated carbohydrates.

## Immunofluorescence

Anti-β-glucan antibody (Abcam, ab233743, dilution 1:100) and Dectin-1-Fc (MedChemExpress, HY-P70101, 10 μg/mL) or CLEC2D-Fc (10 μg/mL) proteins were used to bind with live *C. albicans* yeast cells. After washing, the yeast cells were stained with FITC-labeled goat anti-human IgG1-Fc antibody (Jackson ImmunoResearch, 109-095-190, 1:100) or donkey anti-mouse IgG H&L (Abcam, ab150108, dilution 1:100). The stained particles were then incubated with Calcofluor White and mounted on glass slides for examination using a confocal microscope. Calcofluor White was used to visualize the fungal cell wall, which contains β-glucan, and to confirm the presence of the yeast cells.

$1 \times 10^6$ PBMCs were stimulated with 20 μg curdlan and then fixed with 4% paraformaldehyde. These cells were permeabilized with 0.25% Triton X-100 and incubated with anti-CLEC2D antibody (produced in Jia's lab, 20 μg/mL) at 4 °C for 2 h. Donkey anti-mouse IgG H&L (Alexa Fluor 594) was used as a secondary antibody to detect the location of CLEC2D on the cells. Meanwhile, these cells were stained with FITC-labeled mouse anti-human TLR2 antibody to detect the presence of TLR2. DAPI was used to label the cell nuclei. The fluorescence was detected using a Zeiss LSM880 confocal laser scanning microscope. This experiment investigated the co-localization with CLEC2D and TLR2 in PBMCs with or without curdlan stimulation. Pearson's correlation coefficients (PCCs) were calculated by Plugin "Coloc2" of Image J (version 1.8.0). The protocol was described in chapter 19 3.9 Performing Colocalization Analysis Using Fiji Plugin "Coloc2"[50].

## ChIP-qPCR assay

Chromatin immunoprecipitation (ChIP) assays were performed using a SimpleChIP Enzymatic Chromatin IP Kit from Cell Signaling Technology. Bone marrow-derived dendritic cells (BMDCs) were stimulated with curdlan for 2 h in a 15 cm Petri dish. The cells were then fixed with 1% formaldehyde and lysed to obtain the nucleus. The DNA was digested into fragments ranging from 150–900 base pairs in size using Micrococcal Nuclease. Up to 500 μL of lysate was sonicated to disrupt the nuclear envelope, and the resulting DNA fragments that were associated with transcription factors were immunoprecipitated using specific antibodies against IRF5 or H3, or immunoglobulin G as a negative control. The precipitated DNA fragments were purified using columns and quantified using reverse transcription PCR (RT-PCR) with specific primers listed in Supplemental Table 1 to determine the level of transcription factor binding at particular loci of interest.

## Cytokine measurement

This procedure outlines the steps followed to measure cytokine levels in cell culture supernatant and tissue homogenate samples using an enzyme-linked immunosorbent assay (ELISA) kit from eBioscience. The ELISA plate is coated with a capture antibody specific to the cytokine of interest, which binds to the cytokine in the sample during the overnight incubation. After blocking to prevent non-specific binding, the samples are added to the plates and incubated for 2 h, allowing for any cytokines present in the sample to bind to the capture antibody. HRP-conjugated secondary antibodies are then added, which recognize and bind to the capture antibody-cytokine complex. After washing to remove any unbound antibodies, a substrate solution is added, resulting in a color change that can be detected using a standard instrument that measures absorbance at 450 nm. The OD value is proportional to the amount of cytokine present in the sample.

## Co-immunoprecipitation

HEK293T cells were lysed using Western and IP buffer (Beyotime, P0013) for 15 min. Magnetic beads conjugated with different antibodies were used to immunoprecipitate Flag-tagged proteins, tandem ubiquitin binding entities (TUBEs), or other target proteins including Myc, His, or hTLR2. The samples were incubated overnight at 4 °C, washed with Tween 20/TBS buffer, and eluted with 2× SDS loading buffer for immunoblot assay.

Bone marrow-derived dendritic cells (BMDCs, $1 \times 10^8$) were stimulated with 200 μg/well Curdlan and HK *C. albicans* yeast (MOI = 5) for the indicated time. After stimulation, the cells were lysed with Western and IP buffer, and 800 μL of the cell lysates were incubated with an anti-TLR2 antibody (Abcam, ab209217). Isotype control rabbit IgG was included as a control. Protein A/G magnetic beads were then added to capture immunocomplexes containing TLR2 and its associated proteins. Western blot gels were obtained by Fully Automatic chemiluminescence/fluorescence image analysis System (Tanon 5200s). Image J (version 1.8.0) was used for densitometry analysis.

## Quantitative real-time PCR

RNA was extracted from BMDCs using Trizol and retro-transcribed into cDNA using the PrimeScript RT Master Mix (Takara, RR036A). Real-time PCR was then performed on the cDNA using Power SYBR Green PCR Master Mix and a LightCycler 96 Sequence Detection System. The primer sequences used in the PCR are listed in Supplemental Table 1.

## RNA-seq analysis

Murine BMDCs were stimulated with curdlan (25 μg/well), and BMDCs were collected and frozen in Lysis/Binding Buffer and total RNA extracted using miRVana kit (Invitrogen). The integrity of RNA was assessed using the Agilent 2100 Bioanalyzer (Agilent Technologies, Santa Clara). Then the libraries were constructed using TruSeq Stranded mRNA LT Sample Prep Kit (Illumina). The transcriptome sequencing and analysis were conducted by OE Biotech Co., Ltd. (Shanghai, China), The libraries were sequenced on the Illumina sequencing platform(HiSeqTM 2500 or Illumina HiSeq X Ten).

Differential expression analysis was performed using the DESeq(2012) R package. T-test threshold (p values < 0.05) and fold-change threshold (>1.1 or <0.9) were set as the threshold for significantly differential expression genes (DEGs). GO enrichment and KEGG pathway enrichment analysis of DEGs were performed respectively using R based on the hypergeometric distribution. Heatmapping were performed in R package version1.0.8. Gene set enrichment analysis (GSEA 3.0, Broad Institute) was performed to determine gene sets and pathways that are significantly enriched in DEGs for each group of comparison against the GSEA molecular signature database.

## Bimolecular fluorescence complementation assay

hCLEC2D and hTLR2 were fused to either YFP$^N$ or YFP$^C$, resulting in four different constructs as previously described[9]. These constructs were then transfected into HEK293 cells using lipo2000 (Thermo-Fisher, 11668019), a transfection reagent, and different combinations of constructs were used. The transfected cells were then analyzed using flow cytometry or confocal laser scanning microscope fluorescence microscopy.

## Fluorescence resonance energy transfer (FRET)

20 μg Curdlan in 200 μl Ethanol absolute were coated on 6 well plate overnight. $3 \times 10^6$ PBMCs were treated for 60 min with the presence of Curdlan, and then fixed with 4% paraformaldehyde. Cy5 conjugated TLR2 (Bioss, bs-10472R-Cy5, dilution1:10) and Cy3 conjugated CLEC2D (Bioss, bs-2683R-Cy3, dilution1:10) antibodies were added to fixed cells in a 1:1 mixture of donor/acceptor labeled antibodies. After the 2 h incubation period, cells were washed 3 times with PBS and mounted on microscope slides using antifade reagent (Molecular Probes # P36941). FRET efficiency was assessed using a confocal microscope LSM880 (Zeiss, Germany) with the acceptor photobleaching method.

## Statistics and reproducibility

All experiments were performed at least three times. Immuno-fluorescence staining, immunohistochemical staining, western blot assays, and DNA agarose gel blot representative images are shown. Quantification of all western blots were analyzed by Image J (version 1.8.0). Data were analyzed using GraphPad Prism 8. Unpaired two-tailed Student's $t$ test was used to analyze the differences between two groups. Comparisons among multiple groups were analyzed with one-way ANOVA. Survival curves were compared using the log-rank test. The results are presented as means ± SEM. $P < 0.05$ was considered statistically significant.

## Reporting summary

Further information on research design is available in the Nature Portfolio Reporting Summary linked to this article.

## Data availability

The sequence data generated in this study have been deposited in the GEO database under the accession code PRJNA962091. The data supporting the findings from this study are available within the article file and its supplementary information/source data file. Any other raw data or noncommercial material used in this study are available from the corresponding author upon reasonable request. Source data are provided as a Source Data file. Source data are provided with this paper.

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

## Acknowledgements

This work was supported by the National Key Research and Development Program of China (2022YFC2303000 and 2021YFC2300400 to X.M.J.), the National Natural Science Foundation of China (82371777 and 31970889 to X.M.J.), Innovation Program of Shanghai Municipal Education Commission (201901070007E00022 to X.M.J.), the Key fund for basic research of Shanghai Science and Technology Commission (20JC1417700 to X.M.J.), Shanghai Laboratory Animal Research Fund (21140903300 to X.M.J), Outstanding Research Project of Shanghai Municipal Health Commission (20224Z0020 to X.M.J.) and Sailing Program of Shanghai Science and Technology Commission (22YF1433900 to Y.G.).

## Author contributions

F.L., H.W. and Y.Q.L. performed the experiments. F.L. and H.W. performed the statistical analyses. F.L. and X.M.J. designed the study. F.L., H.W., Y.G. and X.M.J. drafted the manuscript. All authors discussed the results and edited and approved the draft and final versions of the manuscript.

## Competing interests

The authors declare no competing interests.
