## [Peer Review File · Nature Communications]

C-type lectin receptor 2d forms homodimers and heterodimers with TLR2 to negatively regulate IRF5-mediated antifungal immunityReviewers' Comments:

Reviewer #1:

Remarks to the Author:

Overall:

The evidences for the formation of CLEC2d and TLR2 is based on transfection/over-expression. The author needs to use immunoprecipitation show that CLEC2d and TLR2 form heterodimer in primary cells, as overexpression frequently cause non-specific interaction. It will be more convincing if authors can show co-localization of CLEC2d and TLR2 by fluorescence resonance energy transfer(FRET) before cell activation in primary cells. If CLEC2d and TLR2 do not form heterodimer in physiological condition, authors need to demonstrate whether *Candida albicans* is able to induce formation of CLEC2D and TLR2 in primary cells.

In addition, CLEC2d is the ligand of NK receptor NKR-P1B, and is able to downregulate NK functions via engagement of ITIM-containing NKR-P1B. Authors show that overexpression of CLEC2d in RAW cells downregulate IRF-5-mediated signaling, but does not show the mechanism how CLEC2d is able to downregulate the activation of RAW cells. Does NK cells express NKR-P1B, and CLEC2d downregulates RAW cell via cis- or trans-interaction on cell surface? It is important to clarify these points mentioned above.

As CLEC2d itself does not contain signal transduction motif, it is hard to understand how CLEC2d mediates signal transduction after engagement with *Candida albicans* in primary cells.

Specific comments:

Abstract:

line 37-38:

TLR2 has been shown to associate with C-type lectin 5A (CLEC5A), a Syk-coupled C-type lectin, to form heterodimer after engagement with *Listeria monocytogenes*. It is misleading to claim this phenomenon say ' -- ---remain unclear about the possibility and role of dimerization between CLRs and TLRs'.

Line 93-94:

The statement is incorrect and mis-leading. While *Listeria* does engage both CLEC5A and TLR2 to form heterodimer and induce inflammatory reactions (Chen ST et al NC, 2017), dengue virus does not bind TLR2. Instead, dengue virus engages CLEC2 in platelet to induce the release of exosomes and microvesicles, which activate CLEC5A and TLR2, respectively, to enhance NET formation and proinflammatory cytokine release (Sung et al, NC, 2019).

Line 97:

The starting of a sentence should not begin with 'And'.

Line 103-104:

TLR2 has been shown to associate with C-type lectin 5A (CLEC5A), a Syk-coupled CLR, to form heterodimer previously, so it is inappropriate to say ' -- remain unclear about the possibility and role of dimerization between CLRs and TLRs. This sentence is identical to that found in line 37-38.

Figure 1d&e:

It will be more convincing to stain heat-inactivated *Candida albicans* with soluble CLEC2d/TLR2 dimer, and whether it can compete with dectin-1 binding. As primary cells also express dectin-1, author should check whether CLEC2d and TLR2 still forms heterodimer after engagement with *C. albicans* in 293 T cells and in primary cells.

Figure 2 and Supp Fig 2:

Authors should perform competition assay between clec2d and dectin-1, and measure affinity between how CLEC2d mediates signal transduction after engagement with *Candida albicans* in primary cells clec2d and beta-glucan using dectin-1 as control by surface plasmon resonance (SPR) in primary cells.

Discussion

Authors should spend more effort to convince audiences why CLEC2d mediates MyD88 degradation in primary cells, if they do not have data to support this observation, considering CLEC2d itself does not contain motif for signal transduction.

Reviewer #2:

Remarks to the Author:

The manuscript by Li and colleagues analyzed the involvement of the C-type lectin receptor 2d (Clec2d) during candidiasis. Clec2d has been shown to be involved in *Cryptococcus* immunity. Here, the authors found that Clec2d forms dimers with TLR2, binds to beta-glucan and alpha-mannan, and regulates MyD88 degradation to inhibit IRF activation. Clec2d knockout mice were more resistant to infection by enhancing NK cell production. The manuscript is well executed and written. Some points need to be addressed.

Major:

-The manuscript focuses on Clec2d-IRF5 antifungal immunity. However, a description about IRF5 and the context is lacking in the introduction.

-Fig. 2: The authors use heat-killed *Candida* (exposed beta-glucan during heat inactivation) and curdlan (water-insoluble beta-glucan) to show that Clec2d-FC binds this carbohydrate. Does Clec2d binds to exposed beta-glucan on live *Candida* (yeast and hyphae)? Does this recognition competes with Dectin-1? Alternatively, does Clec2d favor mannan on the living organism? An immunofluorescence assay using Dectin-1-Fc and Clec2d-Fc will clarify this.

-Fig.3: The authors use the identified ligand (beta-glucan) to perform downstream signaling analysis. Given that *Candida* is a dynamic organism (yeast to hyphae transition, beta-glucan masking, etc.) some experiments (IRF5 and Myd88 immunoblots) need to be verified with live *C. albicans*.

-Fig. 4: The authors performed microarrays to identify differences in cytokine regulation. Despite the fact that Clec2d BMDCs are highly inflammatory (IL-23, IL-1a/b, IL-6 etc.) compared to WT BMDCs, the authors focused on one identified cytokine, IL-12. Are the other cytokines negatively regulated as well (in vitro and in vivo) since some of these are crucial for antifungal defense, and to prevent immunopathology (PMID: 16619190; PMID: 36138043). In this line, does Clec2d deficiency influences phagocytosis and immune cell survival?

-Fig.5/6: The authors used a mouse model of disseminated candidiasis showing that Clec2d KO mice (and *Lyz2* cre mice) are more resistant to this infection. While mice deficient for Clec2d show increased survival, data presented for fungal burden is exaggerated due to the axis scaling. At first, it looks that these mice clear the fungus completely; however, there is only a slight to modest decrease in burden suggesting tolerance rather than resistance. In line, with fig. 4, the authors show that IL-23 levels increase leading to IFN γ -NK cell recruitment. What happens to the neutrophil population (main antifungal effector cell types during candidiasis) in KO mice? Does NK-cell depletion reduces resistance of Clec2d KO mice?

Minor:

-Does Dectin-1 surface expression changes in Clec2d knockout cells?

Reviewer #3:

Remarks to the Author:

In the manuscript named „C-type lectin receptor 2d forms homodimers and heterodimers with TLR2 to negatively regulate IRF5-mediated antifungal immunity“, the authors show that Clec2d forms homodimers and heterodimers with TLR2. This was confirmed using Coimmunoprecipitation as well as BiFC. The authors pinpoint the dimerization area in Clec2d to the extracellular domain and certain positively charged amino acids. Signalling studies allowed the authors to find that MyD88 degradation and Irf-5 activation were influenced by Clec2d, but not p65 activation or Syk activation. Transcriptome analyses of Clec2d proficient and deficient cells revealed a variety of deregulated pathways upon Clec2d deficiency, of which the authors focused on IL-12 production. Infection experiments of Clec2d deficient mice (complete k.o.) as well as myeloid-specific Clec2d knockout mice revealed a much enhanced resistance of the mice against the fungus, coupled to increased IL-12 production and attraction / activation of NK-cells.

Overall, the manuscript covers experiments from basic biochemistry (interaction studies) via cell biology (signalling studies) to biomedical research (analyses in knockout mice) and is thus definitely a candidate for publication in Nature Communications. However, several critical points need to be addressed to make the paper acceptable:

Major points:

- 1) In all legends, the number of times each experiment was performed needs to be indicated. In some cases, differences are minor and reproducibility is an issue.
- 2) Similarly, all Western Blots shown must be quantified, and graphs showing results of at least 3 independent experiments must be prepared. In several cases, I could not follow the interpretation of the authors without seeing quantification.
- 3) The infection experiments in TLR2/Clec2d double knockout mice are confusing. TLR2 knockout alone has been shown to yield contradictory effects. The authors should compare Clec2d/TLR2 double knockout mice to TLR2 single knockout mice to arrive at sound conclusions. Comparing the double knockout to wt may be misleading. In principle, though, if the authors are unable to provide this comparison, the double knockout might be deleted from the paper without adverse effects.

Minor points:

- 1) In all graphs, the Y axis should start at 0. Cutting the axis as in Figure 1h is not acceptable.
- 2) There are several English mistakes throughout the paper. The authors should consider having it proof-read by a native speaker.

In general, I am not aware of inhibitory dimerization of Toll like receptors or other PRRs. If these examples exist, the authors should state them clearly in the discussion. If not, they should stress their new point more, namely that this is the first inhibitory dimer.

Response to reviewers' comments:

We would like to express our sincere appreciation to the editors and reviewers for the invaluable suggestions and critical comments, which have greatly improved the quality of our manuscript. Based on that, we have performed additional experiments and revised the manuscript accordingly. Here, we present our responses to the reviewer's comments along with the relevant experimental data.

Reviewer #1 (Remarks to the Author):

Overall:

The evidences for the formation of CLEC2d and TLR2 is based on transfection/over-expression. The author needs to use immunoprecipitation show that CLEC2d and TLR2 form heterodimer in primary cells, as overexpression frequently cause non-specific interaction. It will be more convincing if authors can show co-localization of CLEC2d and TLR2 by fluorescence resonance energy transfer (FRET) before cell activation in primary cells. If CLEC2d and TLR2 do not form heterodimer in physiological condition, authors need to demonstrate whether *Candida albicans* is able to induce formation of CLEC2D and TLR2 in primary cells.

Response: We appreciate the reviewer's comment and have conducted additional experiments to evidence this issue. To explore the heterodimerization of CLEC2D and TLR2, we performed immunoprecipitation assay in bone marrow-derived dendritic cells (BMDCs) and found that the association of endogenous CLEC2D and TLR2 in unstimulated BMDCs was weak, whereas the accumulation of endogenous CLEC2D-associated TLR2 protein in BMDCs was significantly increased after stimulation with β -glucan-containing particle curdlan or heat-activated *C. albicans* yeast (Fig.1c-d). We further performed the bimolecular fluorescence complementation (BiFC) assay, which is a technique used to detect protein-protein interactions in living cells. After fusing human CLEC2D and TLR2 proteins to YFP^N or YFP^C, the two fused proteins interacted, causing YFP^N and YFP^C to come into close proximity, resulting in FRET and fluorescence signal. By using flow cytometry or microscopy, we can detect and quantify the level of CLEC2D and TLR2 protein interactions. We found that the combination of YFP^N-CLEC2D and YFP^C-TLR2 formed complexes on the cell surface and emitted fluorescence (Fig. 1h and Supplementary Fig. 1d), implying that CLEC2D formed heterodimers with TLR2. Thus, we performed co-immunoprecipitation

and BiFC assays to provide definitive evidence of a direct interaction of CLEC2D and TLR2. More importantly, the co-localization of endogenous CLEC2D and TLR2 in human peripheral blood mononuclear cells (PBMCs) was also weak in physiological condition (Fig. 1i) whereas the increased co-localization of endogenous CLEC2D and TLR2 in PBMCs after stimulation with β -glucans was observed by immunofluorescence microscopy (Fig. 1i).

In addition, CLEC2d is the ligand of NK receptor NKR-P1B, and is able to downregulate NK functions via engagement of ITIM-containing NKR-P1B. Authors show that overexpression of CLEC2d in RAW cells downregulate IRF-5-mediated signaling, but does not show the mechanism how CLEC2d is able to downregulate the activation of RAW cells. Does NK cells express NKR-P1B, and CLEC2d downregulates RAW cell via cis- or trans-interaction on cell surface? It is important to clarify these points mentioned above.

As CLEC2d itself does not contain signal transduction motif, it is hard to understand how CLEC2d mediates signal transduction after engagement with *Candida albicans* in primary cells.

Response: We would like to express our gratitude to the reviewer for providing valuable insights into the mechanism underlying the downregulation of RAW cell activation mediated by CLEC2d. We show that overexpression of CLEC2D in RAW cells could downregulate IRF5-mediated signaling through promoting Myd88 degradation. NK cells were not involved in this experiment, thus ruling out the possibility that NK cells express NKR-P1B encoding receptor for CLEC2D to enhance CLEC2D-mediated signaling in RAW cells. Moreover, we found that NK1.1 encoded by NKR-P1B gene was undetectable in either RAW cells or BMDCs (Figure R1), which further ruled out another possibility that NKR-P1B expressed in RAW cells or BMDCs to bind CLEC2D for enhancing its signaling.

Regarding to the mechanism how CLEC2D mediated Myd88 degradation and then downregulate IRF5-mediated signaling, we found that inhibition of Smurf1 with its inhibitor Smurf1-IN-A01 completely blocked Clec2d-mediated degradation of Myd88 in RAW264.7 cells after stimulation with β -glucans whereas inhibition of SPOP with its inhibitor SPOP-IN-6b had no influence on Myd88 degradation (Fig. 3n). It has been shown that E3 ligase Smurf1 or SPOP can mediate the ubiquitination and proteasomal degradation of MyD88 in TLR signaling (*Nat Commun* 2011,2,460; *Nat Immunol* 2019, 20, 1196-1207). A previous study shows that Smad6 mediates TGF- β 1-induced K48-linked polyubiquitination and

degradation of Myd88 through recruiting Smurf1 (*Nat Commun* 2011,2,460). Future studies will focus on confirming the hypothesis that CLEC2D may form a complex with Smad6 to recruit Smurf1 for mediating β -glucan-induced K48-linked polyubiquitination and degradation of Myd88.

Figure R1 Flow cytometry analysis of NK1.1 expression on the surface of Raw264.7 cells and BMDCs.

Specific comments:

Abstract:

line 37-38:

TLR2 has been shown to associate with C-type lectin 5A (CLEC5A), a Syk-coupled C-type lectin, to form heterodimer after engagement with *Listeria monocytogenes*. It is misleading to claim this phenomenon say '---remain unclear about the possibility and role of dimerization between CLRs and TLRs'.

Response: Thank you for bringing to our attention the current research progress on TLR2 and CLEC5A during infection with *Listeria monocytogenes*. It has reported that TLR2 and CLEC5A are co-activated by *L. monocytogenes* to enhance host immunity against infection by stimulating the p38- and PI3K-Akt-pathways (*Nat Commun* 2017, 8, 299). However, it has not been proposed or proven whether this co-activation is mediated through heterodimerization between TLR2 and CLEC5A. Based on the current understanding in the field, we decided to keep the original statement in the abstract of our manuscript--- 'it remains unclear about the possibility and role of dimerization between CLRs and TLRs'.

Line 93-94:

The statement is incorrect and mis-leading. While *Listeria* does engage both CLEC5A and TLR2 to form heterodimer and induce inflammatory reactions (Chen ST et al NC, 2017), dengue virus does not bind TLR2. Instead, dengue virus engages CLEC2 in platelet to induce the release

of exosomes and microvesicles, which activate CLEC5A and TLR2, respectively, to enhance NET formation and proinflammatory cytokine release (Sung et al, NC, 2019).

Response: Thank you for drawing our attention to the important series of studies conducted by Chen ST et al. and Sung et al in *Nat. Commun* that explore the role of CLEC5A and TLR2 against *Listeria monocytogenes* and Dengue virus infections. We sincerely apologize for any confusion caused by our previous statements and have taken steps to revise the statement in our revised manuscript to accurately reflect the current state of knowledge in the field-- ‘Furthermore, a study has demonstrated that CLEC5A, DC-SIGN, and mannose receptor (MR) form a multivalent hetero-complex upon engagement with Dengue virus (DV), where DC-SIGN/MR binds DV with high avidity to facilitate CLEC5A-mediated innate responses against DV infection (*PLoS One* 2016, 11, e0166474). Other studies have suggested that DV could activate CLEC2 *via* binding to a DC-SIGN/CLEC2 hetero-multivalent receptor complex in platelets, despite that CLEC2 has low binding affinity to DV (*Nat Commun* 2019, 10, 2402; *Blood* 2013, 121, 95-106). Of note, two recent studies have shown that co-activation of multiple CLEC5A- and TLR2-mediated pathways are required for optimal host innate immunity against infection with DV or *Listeria monocytogenes* (*Nat Commun* 2017, 8, 299; *Nat Commun* 2019, 10, 2402).’

Line 97:

The starting of a sentence should not begin with 'And'.

Response: Thank you for bringing this to our attention. We modified the expression of the sentence and changed the beginning ‘And’ with ‘Of note’. We appreciate your attention to detail and your contribution to enhancing the clarity and professionalism of our manuscript.

Line 103-104:

TLR2 has been shown to associate with C-type lectin 5A (CLEC5A), a Syk-coupled CLR, to form heterodimer previously, so it is inappropriate to say ' -- remain unclear about the possibility and role of dimerization between CLRs and TLRs. This sentence is identical to that found in line 37-38.

Response: Thank you for your thorough review and valuable comments. We would like to extend our sincere apologies for any confusion caused by the previous statements in our

manuscript. Based on the finding that *L. monocytogenes* co-activates TLR2 and CLEC5A to enhance host immunity to infection through the p38 and PI3K-Akt pathways, instead of dimerization between CLR and TLR, we have decided to maintain the original statement in the Introduction section of the manuscript: ‘it remains unclear about the possibility and role of dimerization between TLRs and CLRs’. We appreciate your attention to detail and your efforts to enhance the clarity and professionalism of our work.

Figure 1d&e:

It will be more convincing to stain heat-inactivated *Candida albicans* with soluble CLEC2d/TLR2 dimer, and whether it can compete with dectin-1 binding. As primary cells also express dectin-1, author should check whether CLEC2d and TLR2 still forms heterodimer after engagement with *C. albicans* in 293 T cells and in primary cells.

Response: Thank you for your valuable suggestions. According to your suggestion, we used His-TLR2 as a probe to examine its frequency binding to live *C. albicans* yeast cells with or without combination of Fc-CLEC2D or Fc-Dectin-1 using flow cytometry (Fig. 2f). We found that the addition of Fc-CLEC2D significantly increased the frequency of TLR2 binding to *C. albicans* yeast cells whereas the addition of Fc-Dectin-1 competitively reduced the binding frequency of heterodimeric TLR2/CLEC2D (Fig. 2f). Thus, we suggest that Dectin-1 can compete with CLEC2D/TLR2 heterodimers for binding *C. albicans* yeast cells.

Moreover, we performed immunoprecipitation assay in bone marrow-derived dendritic cells (BMDCs) and found that the association of endogenous CLEC2D and TLR2 in unstimulated BMDCs was weak, whereas the accumulation of endogenous CLEC2D-associated TLR2 protein in BMDCs was significantly increased after stimulation with β -glucan-containing particle curdlan or heat-activated *C. albicans* yeast (Fig.1c-d). We further found that the co-localization of endogenous CLEC2D and TLR2 in human peripheral blood mononuclear cells (PBMCs) was weak in physiological condition (Fig. 1i) whereas the increased co-localization of endogenous CLEC2D and TLR2 in PBMCs after stimulation with β -glucans was observed by immunofluorescence microscopy (Fig. 1i). Thus, these data suggest that CLEC2D and TLR2 interact with each other to form heterodimers in primary cells.

Figure 2 and Supp Fig 2:

Authors should perform competition assay between clec2d and dectin-1, and measure affinity between how CLEC2d mediates signal transduction after engagement with *Candida albicans* in primary cells clec2d and beta-glucan using dectin-1 as control by surface plasmon resonance (SPR) in primary cells.

Response: Thank you for your comments. According to your suggestion, we performed the binding assay of β -glucans with CLEC2D or Dectin-1 as a positive control using immunofluorescence microscopy, which revealed uniform overlaps of CLEC2D or Dectin-1 with β -glucans on the surface of *C. albicans* (Fig. 2b). Moreover, we found that the addition or dilution of His-Dectin-1 protein had no significant influences on the binding affinity of Fc-CLEC2D to β -glucans or the frequency of Fc-CLEC2D binding to *C. albicans* yeast cells (Supplementary Fig. 2c-d).

To rule out the influence of Dectin-1 on CLEC2D-mediated signaling, we overexpressed CLEC2D in RAW 264.7 cells, which did not endogenously express Dectin-1. We found that CLEC2D overexpressed in RAW264.7 cells significantly inhibited β -glucan-induced phosphorylation and nuclear translocation of IRF5, but not p65 phosphorylation (Fig. 3c-d). Moreover, overexpression of CLEC2D in RAW264.7 cells time-dependently increased β -glucan-induced MyD88 degradation whereas pretreatment with proteasome inhibitor MG132 completely inhibited its degradation (Fig. 3f). We further found that inhibition of Smurf1 with its inhibitor Smurf1-IN-A01 completely blocked CLEC2D-mediated degradation of Myd88 in RAW264.7 cells after stimulation with β -glucans whereas inhibition of SPOP with its inhibitor SPOP-IN-6b had no influence on Myd88 degradation (Fig. 3n).

Discussion

Authors should spend more effort to convince audiences why CLEC2d mediates MyD88 degradation in primary cells, if they do not have data to support this observation, considering CLEC2d itself does not contain motif for signal transduction.

Response: Thank you for your valuable suggestions. We revised these content in the section of Discussion as following: ‘Our present study shows that inhibiting Smurf1, but not SPOP, completely blocks Clec2d-mediated degradation of Myd88 induced by β -glucans (Fig. 3n), suggesting that Clec2d promotes MyD88 degradation through recruiting Smurf1. A previous

study had demonstrated that Smad6 mediates TGF- β 1-induced K48-linked polyubiquitination and degradation of Myd88 by recruiting Smurf1 (*Nat Commun* 2011, 2, 460). It has been extensively studied that Smad6 acts as an important mediator to negatively regulate TLR or IL-1R signaling (*Nat Commun* 2011, 2, 460; *Nat Immunol* 2006, 7, 1057-1065). Future studies will focus on confirming the hypothesis that CLEC2D may form a complex with Smad6 to recruit Smurf1 for mediating β -glucan-induced K48-linked polyubiquitination and degradation of Myd88.’

Reviewer #2 (Remarks to the Author):

The manuscript by Li and colleagues analyzed the involvement of the C-type lectin receptor 2d (Clec2d) during candidiasis. Clec2d has been shown to be involved in *Cryptococcus* immunity. Here, the authors found that Clec2d forms dimers with TLR2, binds to beta-glucan and alpha-mannan, and regulates MyD88 degradation to inhibit IRF activation. Clec2d knockout mice were more resistant to infection by enhancing NK cell production. The manuscript is well executed and written. Some points need to be addressed.

Response: Thank you for your review of our manuscript. We appreciate your positive feedback, and we will carefully consider your suggestions for addressing the points that need to be improved. Thank you for your valuable contribution to the peer-review process.

Major:

The manuscript focuses on Clec2d-IRF5 antifungal immunity. However, a description about IRF5 and the context is lacking in the introduction.

Response: We appreciate your suggestion to provide more information about IRF5 in the Introduction section of our revised manuscript as following: “It has been well-documented that IRF5 directly interacts with and is activated by MyD88, functioning as a key transcription factor for TLR-dependent induction of proinflammatory cytokine genes, such as IL-12 (*Nature* 2005, 434, 243-249). Two recent studies have revealed that Dectin-1 activation by *C. albicans* triggered IRF5-mediated expression of interferon (IFN)- β in DCs (*Immunity* 2013, 38, 1176-1186; *Nat Immunol* 2022, 23, 1735-1748).”

Fig. 2: The authors use heat-killed *Candida* (exposed beta-glucan during heat inactivation) and curdlan (water-insoluble beta-glucan) to show that Clec2d-FC binds this carbohydrate. Does Clec2d binds to exposed beta-glucan on live *Candida* (yeast and hyphae)? Does this recognition compete with Dectin-1? Alternatively, does Clec2d favor mannan on the living organism? An immunofluorescence assay using Dectin-1-Fc and Clec2d-Fc will clarify this.

Response: Thank you for your helpful comments. According to your suggestion, we performed the binding assay of β -glucans with CLEC2D or Dectin-1 as a positive control using immunofluorescence microscopy, which revealed uniform overlaps of CLEC2D or Dectin-1 with β -glucans on the surface of *C. albicans* (Fig. 2b). Moreover, we found that the addition or dilution of His-Dectin-1 protein had no significant influences on the binding affinity of Fc-CLEC2D to β -glucans or the frequency of Fc-CLEC2D binding to live *C. albicans* yeast cells (Supplementary Fig. 2c-d).

Fig.3: The authors use the identified ligand (beta-glucan) to perform downstream signaling analysis. Given that *Candida* is a dynamic organism (yeast to hyphae transition, beta-glucan masking, etc.) some experiments (IRF5 and Myd88 immunoblots) need to be verified with live *C. albicans*.

Response: We appreciate your valuable comments and according to your suggestion, we found that the accumulation of endogenous CLEC2D-associated TLR2 protein in bone marrow-derived dendritic cells (BMDCs) was significantly increased after stimulation with β -glucan-containing particle curdlan or heat-activated *C. albicans* yeast (Fig.1c-d). We further found that IRF5 phosphorylation in *Clec2d*-deficient BMDCs was significantly increased after stimulation with *C. albicans* yeast, but not hyphae (Supplementary Fig. 3a-b). We surprisingly found that β -glucan or *C. albicans* yeast stimulation time-dependently promoted MyD88 degradation in BMDCs whereas *Clec2d* deficiency favored its stabilization (Fig. 3e and Supplementary Fig. 3a).

Fig. 4: The authors performed microarrays to identify differences in cytokine regulation. Despite the fact that Clec2d BMDCs are highly inflammatory (IL-23, IL-1a/b, IL-6 etc.) compared to WT BMDCs, the authors focused on one identified cytokine, IL-12. Are the other cytokines negatively regulated as well (in vitro and in vivo) since some of these are crucial for antifungal

defense, and to prevent immunopathology (PMID: 16619190; PMID: 36138043). In this line, does Clec2d deficiency influence phagocytosis and immune cell survival?

Response: Thank you for your valuable suggestions. As shown in Figure 3, *Clec2d* deficiency had no influence on β -glucan-induced phosphorylation of Syk and p65 in BMDCs (Fig. 3a-b). Moreover, *Clec2d* deficiency slightly increased β -glucan-induced phosphorylation of p38, JNK and ERK1/2, which were not affected by the overexpression of Clec2d in RAW264.7 cells (Supplementary Fig. 3c-d). We further found that *Clec2d* deficiency had no influence on β -glucan-induced cytokine production including TNF- α , IL-6 and IL-10, and a slight decrease of IL-1 β production in BMDCs (Figure R2a). Consistently, *Clec2d* deficiency had no influence on the production of TNF- α , IL-6 and IL-1 β in the kidney of *C. albicans*-infected mice (Supplementary Fig. 5a). Together, these data suggested that Clec2d was not involved in the regulation of these pro-inflammatory cytokine production.

We further found that *Clec2d* deficiency had no influence on the phagocytosis of peritoneal macrophages as well as cell survival of BMDCs, which were co-cultured with *C. albicans* (Figure R2b-c).

Figure R2 (a) ELISA results of TNF- α , IL-6, IL-10 and IL-1 β in the supernatants of wild-type and *Clec2d*-deficient BMDCs, which were stimulated with curdlan (5 μ g/well) for the indicated time. **(b)** Phagocytosis assay of peritoneal macrophages from wild-type and *Clec2d*-deficient mice, which were co-cultured with GFP-tagged *C. albicans* at the indicated MOI for 1hour and then analyzed by flow cytometry. **(c)** Survival assay of BMDCs

from wild-type and *Clec2d*-deficient mice, which were stimulated with Curdlan, *C. albicans* yeast, α -mannan and *C. albicans* hyphae for the indicated time and then analyzed by flow cytometry.

Fig.5/6: The authors used a mouse model of disseminated candidiasis showing that *Clec2d* KO mice (and *Lyz2* cre mice) are more resistant to this infection. While mice deficient for *Clec2d* show increased survival, data presented for fungal burden is exaggerated due to the axis scaling. At first, it looks that these mice clear the fungus completely; however, there is only a slight to modest decrease in burden suggesting tolerance rather than resistance. In line, with fig. 4, the authors show that IL-23 levels increase leading to IFN γ +NK cell recruitment. What happens to the neutrophil population (main antifungal effector cell types during candidiasis) in KO mice? Does NK-cell depletion reduces resistance of *Clec2d* KO mice?

Response: Thank you for your valuable suggestions. The findings from the mouse model of disseminated candidiasis suggest that CLEC2D may play a role in susceptibility to this infection. While the use of *Clec2d* KO mice and *Lyz2*^{cre} mice showed increased survival rates compared to wild-type mice, the interpretation of the fungal burden data may be limited by the scaling of the axes. Therefore, we have revised the format of Y axis scaling for showing fungal burden data in Fig. 5b, Fig. 5g, Fig. 5i, Fig. 6b and Fig. 6g of our revised manuscript reference to literature (PMID: 27428901).

As per your suggestion, we examined the neutrophil population in the mouse model of disseminated candidiasis and found that *Clec2d* deficiency resulted in a significant decrease of frequency and numbers of neutrophils in kidneys of *Clec2d*-deficient mice after infection with *C. albicans* for 2 days (Fig. R3). These data ruled out the possibility that *Clec2d* deficiency conferred mice resistance to *C. albicans* infection through increasing frequency and numbers of neutrophils. To determine the role of NK cells in *Clec2d*-deficient mice during *C. albicans* infection, we successfully depleted NK cells using an anti-NK1.1 Ab and found that depletion of NK cells in either wild-type or *Clec2d*-deficient mice led to a higher fungal burden in kidney and brain than those who received control IgG after infection with *C. albicans* (Fig. 5i). These data further confirmed that CLEC2D suppressed host antifungal immunity against *C. albicans* infection through inhibiting the generation of NK cells.

Figure R3 Frequency and numbers of neutrophils in kidneys of wild-type and *Clec2d*-deficient mice after infection with *C. albicans* for indicated time.

Minor:

Does Dectin-1 surface expression changes in *Clec2d* knockout cells?

Response: Thank you for your valuable comments. We examined Dectin-1 surface expression changes in wild-type and *Clec2d*-deficient BMDCs and found that Dectin-1 surface expression slightly increased in *Clec2d*-deficient BMDCs (Fig. R4). To rule out the influence of Dectin-1 on CLEC2D-mediated signaling, we overexpressed CLEC2D in RAW 264.7 cells, which did not endogenously express Dectin-1. We found that CLEC2D overexpressed in RAW264.7 cells significantly inhibited β -glucan-induced phosphorylation and nuclear translocation of IRF5, but not p65 phosphorylation (Fig. 3c-d). Moreover, overexpression of CLEC2D in RAW264.7 cells time-dependently increased β -glucan-induced MyD88 degradation whereas pretreatment with proteasome inhibitor MG132 completely inhibited its degradation (Fig. 3f).

Figure R4 Flow cytometry analysis of Dectin-1 surface expression in BMDCs from WT, *Clec2d*^{-/-}, *Dectin-1*^{-/-} mice.

Reviewer #3 (Remarks to the Author):

In the manuscript named „C-type lectin receptor 2d forms homodimers and heterodimers with TLR2 to negatively regulate IRF5-mediated antifungal immunity“, the authors show that Clec2d forms homodimers and heterodimers with TLR2. This was confirmed using Coimmunoprecipitation as well as BiFC. The authors pinpoint the dimerization area in Clec2d to the extracellular domain and certain positively charged amino acids. Signalling studies allowed the authors to find that MyD88 degradation and Irf-5 activation were influenced by Clec2d, but not p65 activation or Syk activation. Transcriptome analyses of Clec2d proficient and deficient cells revealed a variety of deregulated pathways upon Clec2d deficiency, of which the authors focused on IL-12 production. Infection experiments of Clec2d deficient mice (complete k.o.) as well as myeloid-specific Clec2d knockout mice revealed a much enhanced resistance of the mice against the fungus, coupled to increased Il-12 production and attraction / activation of NK-cells.

Overall, the manuscript covers experiments from basic biochemistry (interaction studies) via cell biology (signalling studies) to biomedical research (analyses in knockout mice) and is thus definitely a candidate for publication in *Nature Communications*. However, several critical points need to be addressed to make the paper acceptable:

Response: Thank you for your valuable feedback on our manuscript. We appreciate your positive comments and recognize the importance of addressing the critical points you raised to make the paper acceptable for publication in *Nature Communications*. We carefully consider your suggestions and revise the manuscript accordingly to improve its quality and impact. Thank you again for your valuable input, and we hope to provide a satisfactory response.

Major points:

In all legends, the number of times each experiment was performed needs to be indicated. In some cases, differences are minor and reproducibility is an issue.

Response: Thank you for your comments. We apologize for the lack of information regarding the number of times each experiment was performed in our legends. We

understand the importance of reproducibility and ensure that all the revised Figure legends include this information. We appreciate your attention to detail and strive to make our research as transparent and reproducible as possible.

Similarly, all Western Blots shown must be quantified, and graphs showing results of at least 3 independent experiments must be prepared. In several cases, I could not follow the interpretation of the authors without seeing quantification.

Response: Thank you for your valuable feedback. We understand that quantification and reproducibility are necessary for a better interpretation of the results, and we have made sure to address these concerns in the revised manuscript. We have quantified all western blot experiments and provided graphs showing results of at least 3 independent repeat experiments. The quantification of at least 3 repeat western blot experiments have also been provided in the Source Data. We apologize for any inconvenience caused by the lack of quantification and reproducibility in the initial submission and thank you for bringing it to our attention.

The infection experiments in TLR2/Clec2d double knockout mice are confusing. TLR2 knockout alone has been shown to yield contradictory effects. The authors should compare Clec2d/TLR2 double knockout mice to TLR2 single knockout mice to arrive at sound conclusions. Comparing the double knockout to wt may be misleading. In principle, though, if the authors are unable to provide this comparison, the double knockout might be deleted from the paper without adverse effects.

Response: Thank you for your helpful suggestions. We have carefully considered your feedback and agree that comparing the Clec2d/TLR2 double knockout mice to wild-type mice may be misleading. To address your concerns, we have removed the data on the double knockout mice from the manuscript. We appreciate your constructive criticism and are grateful for the opportunity to improve our manuscript.

Minor points:

In all graphs, the Y axis should start at 0. Cutting the axis as in Figure 1h is not acceptable.

Response: We appreciate the reviewer's comment regarding the Y-axis starting point on our graphs, especially in Fig. 1h. We apologize for any confusion caused by Y-axis, and we acknowledge the importance of accurate data representation. To ensure that all the graphs in our manuscript accurately reflect our data, we make sure that all the Y-axes start at 0. Thank you for bringing this to our attention, and we do our best to make our research transparent and accurate.

There are several English mistakes throughout the paper. The authors should consider having it proof-read by a native speaker.

Response: Thank you for your valuable comments. We apologize for any English mistakes in our paper and acknowledge the importance of clear and accurate language. We have carefully proofread the grammar and expression of English used in our manuscript. We take quality control seriously and strive to ensure that our research is presented in the best possible light. Thank you again for your constructive criticism and the opportunity to improve our manuscript.

In general, I am not aware of inhibitory dimerization of Toll like receptors or other PRRs. If these examples exist, the authors should state them clearly in the discussion. If not, they should stress their new point more, namely that this is the first inhibitory dimer.

Response: Thank you for your comments. We agree that inhibitory dimerization of Toll-like receptors or other pattern recognition receptors (PRRs) has not been widely reported in the literature. Based on the current understanding in the field, we have made the statement in the Discussion section of our manuscript as following: ‘Traditionally, TLRs and CLRs are considered as stimulatory receptors that upregulate the production of proinflammatory cytokines. Our present data suggest that CLEC2D forms homodimers or heterodimers with TLR2 to function as inhibitory receptors to suppress IRF5-mediated IL-12 production, thereby negatively regulating antifungal immunity against *C. albicans* infection (Fig. 7).’ Our study provides important new insights into the functional diversity of PRR dimerization and activation, specifically the novel finding of inhibitory dimerization of a PRR. We hope that our study will contribute to the growing body of research on the regulation of PRR signaling

and pave the way for future studies in this field. Thank you for your input and bringing this to our attention.

Reviewers' Comments:

Reviewer #1:

Remarks to the Author:

The authors respond to several questions raised in my previous comments, but there is a key issue still not being addressed. The biggest concern of this work is whether CLEC2D and TLR2 can form dimer in primary cells, and the best way to answer this question is to stain the primary cells with fluorochromes-conjugated antibodies to calculate the intensity of FRET. However, the authors circumvent this issue and only perform immunoprecipitation to detect the presence of CLEC2D/TLR2 immune complex from primary cells upon encountering *Candida albicans*. Even the they perform the BiFc assay, it is based on transfection/overexpression system. The author should provide this data to confirm that CLEC2D/TLR2 do associate on cell membrane when encountering *Candida*.

Other comments are as followings:

- 1) Line 118: In the last section of introduction, the author give a short description of their finding. The newly inserted sentences (line 118-122) should be moved to DISCUSSION to support their conclusion regarding the role of IRF5 in their model system.
- 2)Line 136: The author should observe the co-localization of CLEC2D and TLR-2 by immunofluorescence staining, and measure the intensity of Forster Resonance Energy Transfer (FRET). This method is better than immunoprecipitation as the formation of immunocomplex is affected by the stringency of washing condition.
- 3) Line 203: Without measuring the K_a/K_d by SPR or BLI, the imaging can only reveal the co-localization and interaction of two molecules.
- 4) In physiological condition, immune system defend host against *Candida* infection effectively via dectin-1-mediated signaling.
The authors find the presence of CLEC2D/TLR2 negatively regulates TRF-5-mediated anti-fungal immunity. It is hard to understand why healthy host needs this negative regulation pathway during fungal infection? The author should discuss the significance and meaning of their finding from the viewpoint of host defense to fungal infection.

Reviewer #2:

Remarks to the Author:

The authors provided new data in the revised version and addressed all my questions. Congratulations on this work.

Reviewer #3:

Remarks to the Author:

The authors have addressed all my major and minor points satisfactorily. The manuscript is now ready for publication.

Response to reviewers' comments:

We appreciate the comments from the editors and reviewers. We have provided a point-by-point response to the reviewers' comments below, and we hope that our revised manuscript now adequately addresses the concerns raised by reviewers and is suitable for publication in *Nature Communications*.

Reviewer #1 (Remarks to the Author):

The authors respond to several questions raised in my previous comments, but there is a key issue still not being addressed. The biggest concern of this work is whether CLEC2D and TLR2 can form dimer in primary cells, and the best way to answer this question is to stain the primary cells with fluorochromes-conjugated antibodies to calculate the intensity of FRET. However, the authors circumvent this issue and only perform immunoprecipitation to detect the presence of CLEC2D/TLR2 immune complex from primary cells upon encountering *Candida albicans*. Even though they perform the BiFc assay, it is based on transfection/overexpression system. The author should provide this data to confirm that CLEC2D/TLR2 do associate on cell membrane when encountering *Candida*.

Response: Thank you for your comments. Based on your previous comments, we have used immunofluorescence microscopy to provide solid evidence for the association of CLEC2D and TLR2 on the cell membrane of human peripheral blood mononuclear cells (PBMCs). We observed that under physiological conditions, endogenous CLEC2D and TLR2 showed weak colocalization on the cell membrane of PBMCs (Supplementary Fig. 1g). Upon stimulation with curdlan which represent β -glucan components, the colocalization of endogenous CLEC2D and TLR2 in PBMCs was increased (Supplementary Fig. 1g), which corroborated the association of CLEC2D/TLR2 on the cell membrane.

Both Förster resonance energy transfer (FRET) and BiFc are widely used to measure protein-protein interactions in living cells. Using FRET, we conducted further investigations into the interaction between endogenous CLEC2D and TLR2 in human peripheral blood mononuclear cells (PBMCs). Antibodies specific to human CLEC2D and TLR2 were labeled with the fluorescence donor Cy3 and acceptor Cy5, respectively. Following stimulation with β -glucans, we observed a significant increase in FRET efficiency, indicating a rapid

association between CLEC2D and TLR2 on the PBMC membranes (Fig.1i and Supplementary Fig. 1h).

Other comments are as followings:

1) Line 118: In the last section of introduction, the author give a short description of their finding. The newly inserted sentences (line 118-122) should be moved to DISCUSSION to support their conclusion regarding the role of IRF5 in their model system.

Response: We appreciate your suggestion to move the newly inserted sentences (lines 118-122) to the DISCUSSION section to support our conclusion regarding the role of IRF5 in our model system. We agree that these sentences would fit better in the DISCUSSION section, as the last section of INTRODUCTION provide a summary of our findings and their implications. We will make this change in the revised manuscript to improve the overall flow and coherence of the manuscript.

2)Line 136: The author should observe the co-localization of CLEC2D and TLR-2 by immunofluorescence staining, and measure the intensity of Forster Resonance Energy Transfer (FRET). This method is better than immunoprecipitation as the formation of immunocomplex is affected by the stringency of washing condition.

Response: Thank you for discussing the FRET method with us again. As mentioned in the above response, to address the issue of dimer formation of CLEC2D and TLR2 in primary cells, we have already used immunofluorescence microscopy in our revised manuscript, as you suggested. We utilized human peripheral blood mononuclear cells (PBMCs) and observed an increase in the colocalization of endogenous CLEC2D and TLR2 in PBMCs upon stimulation with β -glucans (Supplementary Fig. 1g).

Using Fluorescence Resonance Energy Transfer (FRET), we conducted further investigations into the interaction between endogenous CLEC2D and TLR2 in human peripheral blood mononuclear cells (PBMCs). Antibodies specific to human CLEC2D and TLR2 were labeled with the fluorescence donor Cy3 and acceptor Cy5, respectively. Following stimulation with β -glucans, we observed a significant increase in FRET efficiency, indicating a rapid association between CLEC2D and TLR2 on the PBMC membranes (Fig.1i and Supplementary Fig. 1h).

3) Line 203: Without measuring the K_a/K_d by SPR or BLI, the imaging can only reveal the co-localization and interaction of two molecules.

Response: In Fig. 2d, we performed ELISA assay to examine the binding ability of β -glucans, α -mannans or Pam3CSK4 to the extracellular domain of Fc-fused CLEC2D or His-fused TLR2 (alone or combination). It has been well-documented that ELISA is a conventional experimental method to analyze the binding activity of ligands and receptors. Of note, SPR or BLI cannot be used to measure the K_a/K_d of polysaccharides binding to receptors due to the lack of mature labeling technology.

4) In physiological condition, immune system defend host against *Candida* infection effectively via dectin-1-mediated signaling.

The authors find the presence of CLEC2D/TLR2 negatively regulates IRF-5-mediated anti-fungal immunity. It is hard to understand why healthy host needs this negative regulation pathway during fungal infection? The author should discuss the significance and meaning of their finding from the viewpoint of host defense to fungal infection.

Response: Thank you for your insightful suggestions. It has been shown that Galectin-3 negatively regulates Dectin-1-mediated IL-23 production in DCs to suppress Th17 polarization against *C. albicans* infection and CLEC-1 negatively regulates Dectin-1-mediated IL-1 β production and neutrophils recruitment in response to *Aspergillus fumigatus* infection (*Am J Pathol.* 2013, 183(4):1209-1222; *Invest Ophthalmol Vis Sci.* 2021, 62(6):28). Based on published findings and the present observations, we propose that CLEC2D forms homodimers or heterodimers with TLR2 to function as inhibitory receptors to suppress IRF5-mediated IL-12 production, thereby negatively regulating antifungal immunity against *C. albicans* infection. These insights on homo- or hetero-dimers of CLEC2D and TLR2 enrich the complex landscape of receptor dimerization to regulate host innate immunity against various microbial infections. This may be useful in the development of new immunotherapeutic strategies to combat these devastating infections.

Reviewer #2 (Remarks to the Author):

The authors provided new data in the revised version and addressed all my questions. Congratulations on this work.

Response: Thank you for your positive feedback on our revised manuscript. We appreciate your valuable comments and suggestions, which have contributed to improving the quality of our work. We are glad that the new data we provided have addressed your concerns, and we are grateful for your recognition of our research.

Reviewer #3 (Remarks to the Author):

The authors have addressed all my major and minor points satisfactorily. The manuscript is now ready for publication.

Response: Thank you for your affirmative assessment of our revised manuscript. We appreciate your valuable input and guidance during the review process, which have contributed to enhancing the quality of our work. We are pleased that all your concerns have been resolved satisfactorily, and we are grateful for your acknowledgment of our research. We anticipate the publication of our manuscript.

Reviewers' Comments:

Reviewer #1:

Remarks to the Author:

The authors have addressed my concerns, and I am satisfied with the revised version.

POINT BY POINT REPLY TO THE REVIEWERS' COMMENTS (reproduced verbatim)

Reviewer #1 (Remarks to the Author):

The authors have addressed my concerns, and I am satisfied with the revised version.

Response: We are glad to know that Reviewer #1 had no further concerns and were satisfied with our efforts to address all questions.